# Ribosomal mutations promote the evolution of antibiotic resistance in a multidrug environment

James E Gomez[1†], Benjamin B Kaufmann-Malaga[1,2,3†], Carl N Wivagg[1,2], Peter B Kim[1], Melanie R Silvis[1], Nikolai Renedo[1], Thomas R Ioerger[4], Rushdy Ahmad[1], Jonathan Livny[1], Skye Fishbein[5], James C Sacchettini[6], Steven A Carr[1], Deborah T Hung[1,2,7]*

[1]The Broad Institute of MIT and Harvard, Cambridge, United States; [2]Department of Molecular Biology and Center for Computational and Integrative Biology, Massachusetts General Hospital, Boston, United States; [3]Department of Microbiology and Immunobiology, Harvard Medical School, Boston, United States; [4]Department of Computer Science, Texas A&M University, College Station, United States; [5]Department of Immunology and Infectious Diseases, Harvard T.H. Chan School of Public Health, Boston, United States; [6]Department of Biochemistry and Biophysics, Texas A&M University, College Station, United States; [7]Department of Genetics, Harvard Medical School, Boston, United States

*For correspondence: hung@
molbio.mgh.harvard.edu

†These authors contributed
equally to this work

Competing interests: The
authors declare that no
competing interests exist.

Reviewing editor: Michael S
Gilmore, Harvard Medical
School, United States

**Abstract** Antibiotic resistance arising via chromosomal mutations is typically specific to a particular antibiotic or class of antibiotics. We have identified mutations in genes encoding ribosomal components in *Mycobacterium smegmatis* that confer resistance to several structurally and mechanistically unrelated classes of antibiotics and enhance survival following heat shock and membrane stress. These mutations affect ribosome assembly and cause large-scale transcriptomic and proteomic changes, including the downregulation of the catalase KatG, an activating enzyme required for isoniazid sensitivity, and upregulation of WhiB7, a transcription factor involved in innate antibiotic resistance. Importantly, while these ribosomal mutations have a fitness cost in antibiotic-free medium, in a multidrug environment they promote the evolution of high-level, target-based resistance. Further, suppressor mutations can then be easily acquired to restore wild-type growth. Thus, ribosomal mutations can serve as stepping-stones in an evolutionary path leading to the emergence of high-level, multidrug resistance.

## Introduction

Antibiotic resistance is on the rise in virtually all clinically relevant pathogens. A 2013 report from the CDC (*Centers for Disease Control and Prevention (CDC), 2013*) identified the most significant antibiotic resistant pathogens, including *S. pneumoniae*, extended-spectrum *β*-lactamase producing *Enterobacteriaceae*, vancomycin-resistant Enterococci, and methicillin-resistant *Staphylococcus aureus* (MRSA), which has now achieved pandemic spread (*Monecke et al., 2011*). This report estimates that in the United States, antibiotic resistant infections kill at least 23,000 people each year. The threat of antibiotic resistant bacteria is however not limited to the developing world, as the prevalence of multidrug resistant (MDR) and extensively drug resistant (XDR) tuberculosis (TB) has been on the rise for the last decade (*Migliori et al., 2010*), and untreatable gonorrhoea infections are now emerging (*Unemo and Nicholas, 2012*). A U.K. task force estimates up to 10 million deaths

**eLife digest** The rise of antibiotic resistant bacteria is challenging clinicians, and some infections are now resistant to almost all of the drugs that are currently available. Some types of bacteria – such as mycobacteria, which include the bacteria that cause tuberculosis and leprosy – can only acquire antibiotic resistance from mutations that alter their existing genes. The process by which bacteria develop resistance to multiple drugs is generally viewed as a stepwise accumulation of different mutations. However, the role of individual mutations that increase a bacterium's resistance to multiple antibiotics has not been fully explored.

Gomez, Kaufmann-Malaga et al. exposed bacteria from the species *Mycobacterium smegmatis*, a cousin of the bacterium that causes tuberculosis, to a mixture of relatively low concentrations of different antibiotics that should kill the bacteria relatively slowly. Hundreds of small bacteria cultures were grown in parallel, and only a fraction of them developed antibiotic-resistant members. Gomez, Kaufmann-Malaga et al. identified mutations in these bacteria that unexpectedly gave the bacteria resistance to several unrelated classes of antibiotics.

Individual mutants carried single mutations in different components of the ribosome, a complex molecular machine that helps to build proteins inside cells. As well as increasing their resistance to antibiotics, these mutations also reduced the growth rate of the bacteria. This meant that when the bacteria were grown in an antibiotic-free environment they survived less well than non-mutant bacteria. However, the mutations gave the bacteria an advantage in environments that contained many different antibiotics, as they could more easily develop mutations that made them more resistant to other drugs. Thus, the mutant bacteria can serve as stepping-stones toward the development of high-level resistance to multiple drugs.

Further work will now explore whether this phenomenon occurs in a range of other bacterial species, including the bacteria that cause tuberculosis. While new antibiotics are desperately needed, a better understanding of how bacteria evolve the ability to resist the effects of antibiotics will help us to preserve the usefulness of existing and future drugs.

each year due to antibiotic resistant infections world-wide by 2050 (*O'Neill, 2016*). A better understanding of the events leading to the generation and fixation of resistance-conferring mutations is vital to preserving the effectiveness of current and future antibiotics.

Antibiotic resistance can be acquired by horizontal gene transfer or mutation of existing genes. Mobile elements often carry genes encoding inactivating enzymes (*Ramirez and Tolmasky, 2010*; *Bush, 2013*), efflux systems (*Poole, 2005*), or alternative enzymes that bypass the native enzyme targets of the antibiotic, as is seen with MRSA and vancomycin resistant *Enterococcus* (*de Lencastre et al., 1994*; *Courvalin, 2006*; *Wellington et al., 2013*). However, some bacteria do not readily acquire new DNA, including pathogenic mycobacteria (*Musser, 1995*). Resistance can nonetheless emerge through acquisition of chromosomal mutations that confer resistance in a variety of ways, including altering the target to prevent antibiotic binding (*Musser, 1995*; *Jacoby, 2005*), increasing target expression (*Banerjee et al., 1994*; *Rouse et al., 1995*), decreasing intracellular drug concentration via enhanced efflux or reduced permeability (*Fernández and Hancock, 2012*), or reducing the activation of prodrugs (*Scorpio and Zhang, 1996*; *Zhang et al., 1992*). With these many mutational pathways to resistance available, it is imperative to understand the factors that contribute to the de novo development of antibiotic resistance.

Clinically, the emergence of antibiotic resistance via mutation can be reduced by ensuring that tissue concentrations of antibiotic always exceed a threshold known as the mutant prevention concentration (MPC) (*Zhao and Drlica, 2001*; *Martinez et al., 2012*; *Drusano, 2004*; *Baquero and Negri, 1997*). Above this level, no single mutation can decrease antibiotic sensitivity sufficiently to allow bacterial growth or survival. However, in patients, such concentrations can be difficult to achieve and maintain due to pharmacokinetic and toxicity issues. Concentrations below the MPC but above the minimum inhibitory concentration (MIC) define the traditional mutant selection window (*Drlica, 2003*), the range of concentrations in which a single mutation can confer a selective advantage. However, even concentrations well below the MIC can select for resistant organisms

(*Gullberg et al., 2011*; *Liu et al., 2011*), and the enhanced fitness of even low-level resistant mutants can contribute to the development of high-level resistance (*Baquero et al., 1998*).

The use of antibiotics in combination can also limit the emergence of resistance (*Mouton, 1999*) by requiring a bacterium to acquire multiple mutations simultaneously in order to survive in a multi-drug environment (*Lipsitch and Levin, 1997*; *Fischbach, 2011*). However, multi-antibiotic therapy can be undermined by issues including poor compliance, low quality antibiotics, and inadequate susceptibility data (*Ormerod, 2005*). Combinations of antibiotics with different pharmacokinetic properties, along with pharmacogenomic differences between individuals, can also result in periods where one or more of the antibiotics is present at subinhibitory concentrations, wherein a single mutation could allow for an enrichment of a monoresistant strain, which would in turn increase the likelihood of the emergence of a strain carrying multiple, independent resistance-conferring mutations (*Mitchison, 1998*; *Ramachandran and Swaminathan, 2012*). Also, in many infections, highly dynamic bacterial populations can reach high numbers (>$10^{10}$ organisms) and thus the likelihood of a bacterial lineage becoming MDR during the course of a single human infection may be more common than had been previously perceived (*Pasipanodya and Gumbo, 2011*; *Colijn et al., 2011*).

Antibiotic stress itself can also impact rates of mutation. Increased mutational frequencies can be a direct result of antibiotic action, as seen with the DNA damaging effects of fluoroquinolones (*Cirz et al., 2005*; *Cirz and Romesberg, 2006*), or due to downstream effects of antibiotics, which have been suggested to involve free radical production resulting in damage to the bacterial chromosome (*Kohanski et al., 2010*; *Foti et al., 2012*; *Kohanski et al., 2007*; *Dwyer et al., 2014*). While DNA damage may contribute to the bactericidal activity of antibiotics, it may also promote resistance-conferring mutations. Recently, several investigators have shown mutation rates can be elevated even by exposure to subinhibitory levels of antibiotic (*Monecke et al., 2011*; *Kohanski et al., 2010*; *Blázquez et al., 2012*; *Andersson and Hughes, 2012*).

To gain insight into the molecular processes that promote antibiotic resistance, we investigated the relationships between antibiotic concentration, bactericidal activity, and the emergence of resistant mutants in a model species in which resistance arises nearly exclusively through mutation, *Mycobacterium smegmatis*. We found novel ribosomal mutations that confer resistance to antibiotics of several different classes that do not target the ribosome, as well as to non-antibiotic stressors. Despite the fact that these mutations have an associated fitness cost in the absence of antibiotic stress and confer relatively modest levels of resistance, they can facilitate evolution to high levels of antibiotic resistance while escaping the fitness cost associated with these mutations through a subsequent reversion. Such transient mutations can thus serve as stepping-stones that facilitate evolution from antibiotic sensitivity to multidrug resistance.

## Results

### Selection for low level ciprofloxacin resistance identifies ribosomal mutants

We developed a microtiter-based liquid culture system that enables the isolation of individual antibiotic-resistant mutants while simultaneously monitoring the growth and kill kinetics of the bulk, susceptible population (*Figure 1A*). We uniformly seeded 384-well plates with green fluorescent protein (GFP)-expressing *M. smegmatis* (10 bacilli per well), allowed the bacterial populations in each well to expand in parallel for several generations in the manner of Luria and Delbruck (*Luria and Delbrück, 1943*), and then exposed them to antibiotics. At the start of antibiotic exposure, the per-well population was sufficiently small so that fluorescent signal in each well was below the threshold of detection, which is approximately $10^5$ cells/well (*Figure 1—figure supplement 1*). Over time, fluorescent signal appeared in a fraction of wells due to the outgrowth of an antibiotic-resistant mutant. At regular intervals during antibiotic exposure, eight wells were sacrificed to assess bacterial viability using a most probable number (MPN) assay in order to measure the kinetics of bacterial killing in response to antibiotic treatment (*Figure 1—figure supplement 1*). This sampling showed an initial expansion of the bacterial population to approximately 300 cells per well by 12 hr, followed by a rapid decline during the next 36 hr.

The relatively slow rate of killing by fluoroquinolone (FQ) antibiotics at levels just above the minimum inhibitory concentration (MIC) allows for a prolonged survival time under antibiotic stress,

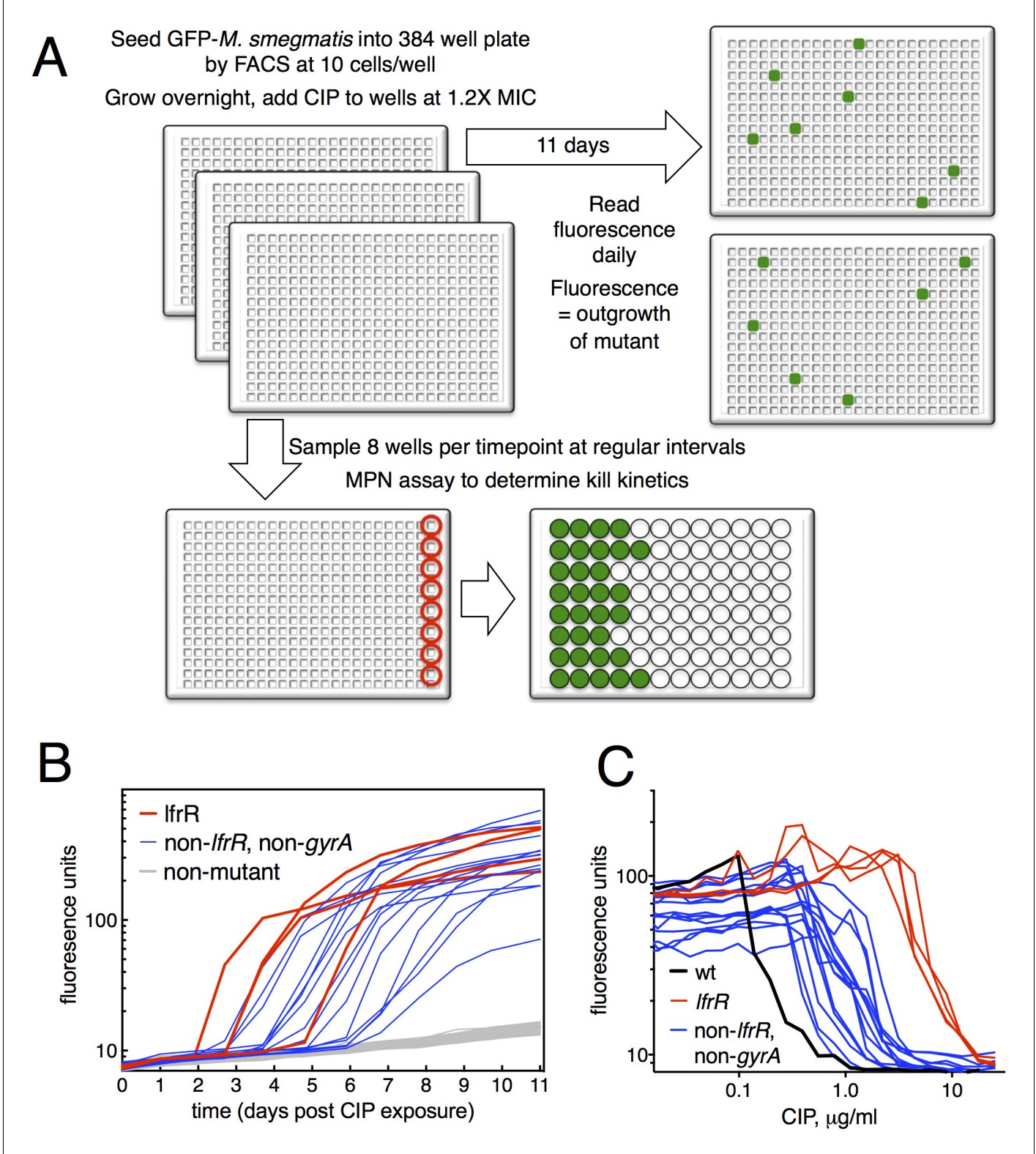

**Figure 1.** Selection of mutants at low ciprofloxacin concentration in 384 well plates. (**A**) Schematic of the selection strategy. Starting at t = −16 hr, 10 *M. smegmatis* cells are introduced into wells of three 384 well microtiter plates containing 15 μl of growth medium and grown overnight at 37°C. At t = 0, 15 μl of medium containing CIP are added to yield a final CIP concentration of 0.3 μg/ml. Plates were sealed with foil and placed at 37°C. At t = 0 and at roughly daily intervals afterwards, plates 1 and 2 were read in a fluorimeter (485 ex., 538 em., 530 cutoff). Plate three was used to monitor bacterial

*Figure 1 continued on next page*

*Figure 1 continued*

survival; at regular intervals, the foil seal was cut and peeled to expose a subset of 8 wells that were harvested using a robotic pipettor and serially diluted (10-fold). Growth of serial dilutions was measured after one week and the bacterial population at the time of harvest was then calculated using most-probable number method. Frequency of resistant mutants is calculated by the equation:$1 - \left(1 - \frac{\text{positive wells}}{\text{total wells}}\right)^{\frac{1}{\text{cells/well}}}$. (B) Emergence of mutants in 0.3 µg/ml CIP. Outgrowth was detected by fluorescence (y-axis). Each line represents a single well followed over time (x-axis). (C) CIP sensitivity of wild-type *M. smegmatis* (black) compared with *lfrR* mutants (red) and non-*lfrR*, non-*gyrA* mutants (blue). Each line represents a unique mutant and illustrates outgrowth as measured by fluorescence (y-axis) as a function of CIP concentration (x-axis). Data are the average of duplicate wells.

The following figure supplements are available for figure 1:

**Figure supplement 1.** Survival of overall bacterial population during selection and determination of limit of detection for outgrowth of a well.

**Figure supplement 2.** Ribosomal mutants can be isolated from a true wild-type *M. smegmatis* strain (mc²6), demonstrating that the fluoroquinolone resistance of ribosomal mutants is not linked to the highly transformable mutant phenotype of mc²155.

providing an enhanced opportunity for the acquisition of mutations. When we exposed *M. smegmatis* to the FQ ciprofloxacin (CIP) at 0.3 µg/ml (1.2X MIC), we observed the emergence of fluorescent signal in 19 of 768 wells, monitoring fluorescence at daily intervals (*Figure 1B*). This fluorescence was due to the outgrowth of CIP resistant *M. smegmatis*; we have confirmed that, in this system, the lack of signal within a well is due to the CIP-dependent killing of CIP-sensitive bacteria and not to plasmid loss in the mc²155 strain, which has been shown to have altered plasmid maintenance (*Panas et al., 2014*) (*Figure 1—figure supplement 2*). This yields a mutation frequency of $9.7 \times 10^{-5}$ based on the peak population of approximately $10^5$ bacteria per 384 well plate seen at 12 hr after introduction of CIP (*Figure 1—figure supplement 1B*), prior to the onset of killing. We measured resistance levels of clones cultured from each of the 19 mutants (as IC50, see Materials and methods), and found IC50 increases ranging from a 2.2 to 44-fold relative to wild-type (*Figure 1C*).

The target of FQs is the A subunit of DNA gyrase, encoded by *gyrA*. Mutations here occur at a frequency of approximately $10^{-7}$ and cause a 12–20 fold increase in MIC to CIP (*Zhou et al., 2000*). We thus PCR amplified and sequenced *gyrA* from each of the 19 isolated mutants, but none had mutations at this locus. We then performed whole genome sequencing (WGS, *Table 1*) of each mutant, which revealed resistance-conferring mutations in the *lfrR* gene in four mutants (21%). The *lfrR* gene encodes a transcriptional repressor of the LfrA efflux pump (*Buroni et al., 2006*). LfrA contributes to the intrinsic resistance of *M. smegmatis* to FQs (*Takiff et al., 1996*; *Sander et al., 2000*), and de-repression of *lfrA* was shown to increase the MIC to CIP 16-fold (*Drusano, 2004*). The isolated *lfrR* mutants were significantly more resistant to CIP (mean 34-fold increase in IC50 over wild-type, *Figure 1C*), than the other 15 mutants (mean 7-fold increase, p=0.0003). The *lfrR* mutants also arose more rapidly, with 3/4 reaching detectable fluorescence levels within 3 days. In contrast, the 15 non-*gyrA*, non-*lfrR* mutants arose more slowly, taking up to 7 days to reach detectable fluorescence (*Figure 1B*).

When we examined the genome sequence of the 15 non-*gyrA*, non-*lfrR* mutants, 14/15 had mutations within genes associated with the ribosome. Ten contained small insertions or deletions in operons encoding ribosomal proteins (*Figure 2A*, *Table 1*), including seven with frameshift mutations in proteins composing the large ribosomal subunit: 5 in *rplO* (L15), 1 in *rplF* (L6), and 1 in *rplY* (L25). One mutant had an intergenic deletion between *rplW* and *rplB* (L23/L2), and two had frameshifts in *rpsE*, encoding the S5 protein of the small subunit. Four of the five remaining mutants had SNPs in *rrlB* (23S rRNA), one of two ribosomal RNA operons in *M. smegmatis* (*Sander et al., 1996*). We were unable to identify any causal mutations in the final mutant using WGS. We confirmed that these ribosomal mutations were directly responsible for FQ-resistant phenotype by engineering selected mutations (*rplO*-1 and *rpsE*-1; see *Table 1*) into the chromosome of *M. smegmatis* using allelic exchange (*Figure 2B*) and measuring the corresponding MICs to CIP. Complementation by a plasmid expressing a wild-type *rplO* allele restored CIP sensitivity to the *rplO*-1 mutant (*Figure 2—figure supplement 1*).

**Table 1.** Genotypes of mutants from primary screen. Polymorphisms relative to *M. smegmatis* mc²155. Coordinates based on position in open reading frame except for B-P7, where genome position is used.

| Mutant | Allele | Gene(s) Mutated | Consequence |
|---|---|---|---|
| A-A13 | rplO-1 | *rplO* 82ΔT (L15) | L15 frameshift @ AA 28 of 147 |
| A-A15 | rplO-2 | *rplO* 77ΔA (L15)<br>*MSMEG_3872* 398ΔTC | L15 frameshift @ AA 26 of 147<br>Precorrin-8X methylmutase frameshift AA 133 0f 208 |
| A-G16 | rrlB-2 | *rrlB* G2296T | non-coding, 23S rRNA |
| A-G20 | rrlB-4/lfrR-3 | *rrlB* C904A<br>*lfrR* T452C | non-coding, 23S rRNA<br>LfrR F151S |
| A-H2 | lfrR-1 | *lfrR* C31A | LfrR R11S |
| A-I6 | lfrR-4 | *lfrR* 63+C | LfrR frameshift @ AA 22 of 189 |
| A-K20 | rpsE-1 | *rpsE* 599-606ΔAGAGTGAA | S6 frameshift @ AA 200 OF 214 |
| A-K24 | rplO-3 | *rplO* 74-77ΔGTGA | L15 frameshift @ AA 26 of 147 |
| A-L8 | rrlB-1 | *rrlB* G2199T | non-coding, 23S rRNA |
| A-M4 | rplO-2 | *rplO* 77ΔA | L15 frameshift @ AA 26 of 147 |
| A-M17 | rplF-1 | *rplF* 481-482ΔAA | L6 frameshift @ AA 161 of 179 |
| B-A9 | lfrR-2 | *lfrR* C98A | LfrR A33E |
| B-I24 | rplO-3 | *rplO* 74-77ΔGTGA | L15 frameshift @ AA 26 of 147 |
| B-O3 | rplY-1 | *rplY* 45ΔG | L25 frameshift @ AA 16 of 215 |
| B-O11 | rpsE-2 | *rpsE* 630+GC | S6 frameshift @ AA 210 OF 214 |
| B-O18 | rrlB-3 | *rrlB* G2891C | non-coding, 23S rRNA |
| B-P7 | rplW/B-1 | 1539202ΔGCAGAGA* | intergenic |
| B-P17 | rrlB-3 | *rrlB* G2891C | non-coding, 23S rRNA |

## Ribosomal mutants are resistant to multiple antibiotics and other stresses

We tested sensitivity to nine additional antimicrobials, hypothesizing that mutations in the translational machinery might have broad effects on innate antibiotic sensitivity given their unexpected impact on FQ sensitivity. Intriguingly, ribosomal mutants showed cross-resistance to antibiotics of unrelated classes to which they had never been exposed (*Figure 2C*), as well as collateral sensitivity to meropenem. The most pronounced shifts were to the cell-wall biosynthesis inhibitor isoniazid (INH), with an average 40-fold increase in IC50 over wild-type (range 6.5 to 158-fold). In contrast, *lfrR* mutants showed little cross-resistance to INH, consistent with previous characterizations of this efflux system. Cross-resistances were not due to acquisition of additional mutations during the MIC assay, because the assay duration does not allow single cells, i.e. de novo mutants, to reach detectable levels of fluorescence. The rplO-1 and rpsE-1 allelic exchange mutants also displayed these phenotypes (*Figure 2—figure supplement 1*). We also evaluated the ability of ribosomal mutants to tolerate non-antibiotic stresses. Ribosomal mutants were better able to tolerate membrane stress due to SDS exposure (*Figure 2D*) and high temperature (54°C, *Figure 2E*) compared to wild-type bacteria. Altogether, these data demonstrate that perturbations to the ribosome can have broad ranging consequences on bacterial survival under duress, leading us to investigate the mechanisms underlying these phenotypes.

The late emergence of ribosomal mutants during CIP selection (*Figure 1A*) suggested that they might have growth defects. We thus first measured doubling times of the ribosomal mutants both in the presence and absence of CIP (*Figure 3A*). In contrast to the 2.4 hr doubling time calculated for the parental wild-type strain or *lfrR* mutants, ribosomal mutants had doubling times ranging from 4–6.5 hr. The slow growth of mutants generated via allelic exchange confirmed that the ribosomal lesions directly reduce growth rates (*Figure 3—figure supplement 1A*). Because conditions that limit bacterial growth have been show to reduce the bactericidal activity of antibiotics (*Tuomanen et al., 1986*), we examined whether the observed shifts in antibiotic sensitivity could be

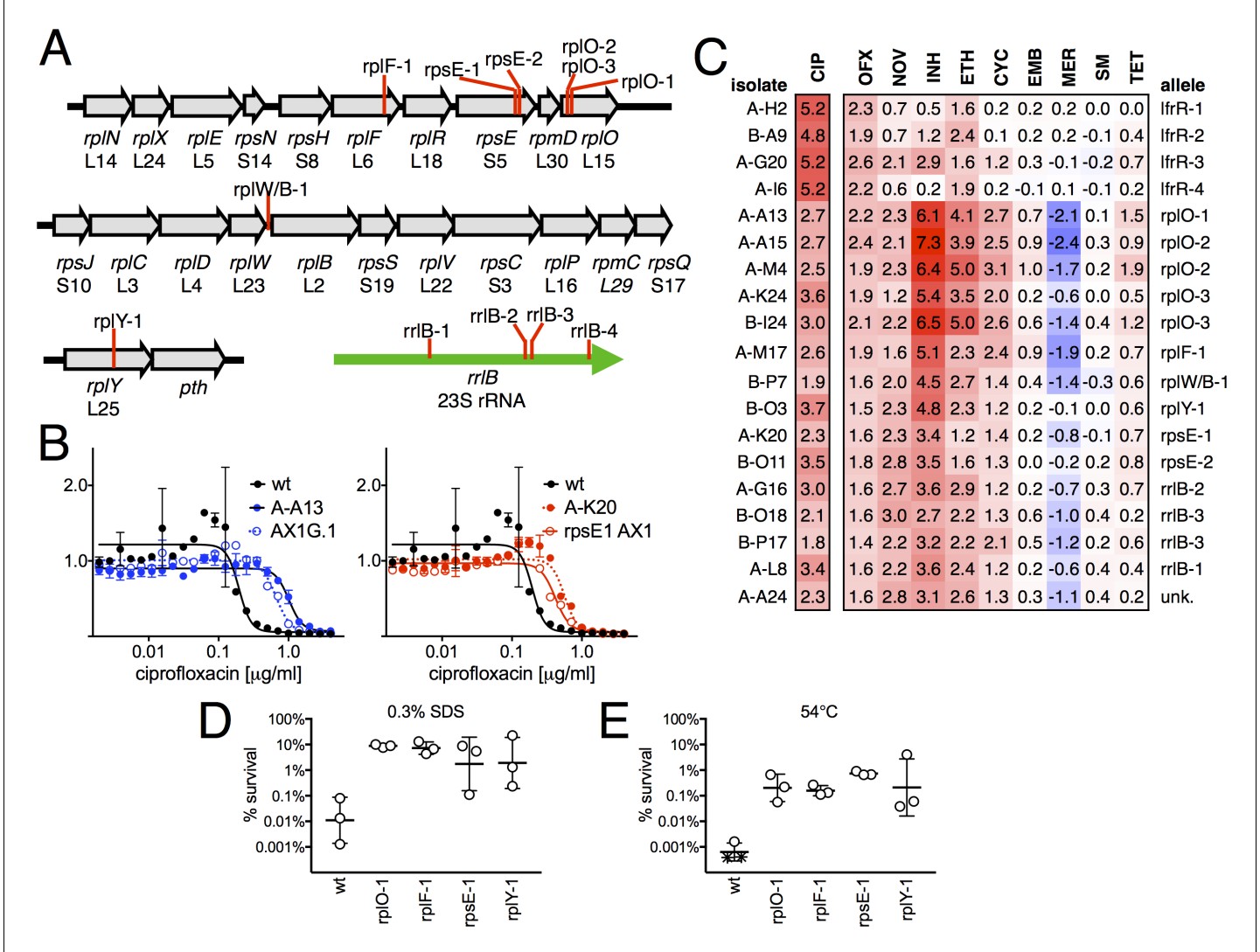

**Figure 2.** Ribosomal mutants are resistant to multiple antibiotics and environmental stressors. (A) Schematic of the four regions of the *M. smegmatis* chromosome in which mutations in ribosomal components were found. Detailed descriptions of the mutations are provided in *Table 1*. (B) Introduction of the rplO-1 or rpsE-1 allele via homologous recombination (yielding AX1G.1 and rpsE1 AX1, respectively) confirms these mutations confer ciprofloxacin resistance. All strains carry pUV3583c and express GFP. Fluorescence of wild-type, mutant, and allelic-exchange derived strains was measured after 2 days of antibiotic exposure. Y axis is GFP signal normalized to untreated controls, x-axis = 1.41 fold ($\sqrt{2}$) dilution series of antibiotic, data are average of duplicate wells. (C) Numbers and shading show the IC50 shift (log2 transformed) of 19 mutants to a panel of antibiotics. Red = resistance, blue = sensitivity. OFX = ofloxacin, NOV = novobiocin, INH = isoniazid, ETH = ethionamide, CYC = cycloserine, EMB = ethambutol, MER = meropenem, SM = streptomycin, TET = tetracycline. (D) Bacteria were exposed to 0.3% SDS for 1 hr, and survival was measured by plating for colony forming units.. Dots represent individual biological replicates, each assayed in duplicate. * indicates < level of detection (approximately 0.0005%). For all mutants tested, <0.05 relative to wt (student's T-test, survival data log transformed). Lines represent mean ± SE. For rpsE-1, p=0.054 relative to wt (student's T-test, survival data log transformed), all others p<0.05. (E) Bacteria were incubated at 54°C for 2 hr. Data represented and analyzed as in (C).

The following figure supplement is available for figure 2:

**Figure supplement 1.** MICs of rplO-1 and rpsE-1 allelic exchange mutants.

a simple consequence of slowed bacterial growth. To directly test whether slow growth itself was causing the MIC shifts, we manipulated the growth rate of wild-type *M. smegmatis* by altering the temperature or restricting the carbon source in the growth medium (*Figure 3B*, *Figure 3—figure supplement 1B*). Slower growth rates alone did not recapitulate the IC50 shifts seen in the

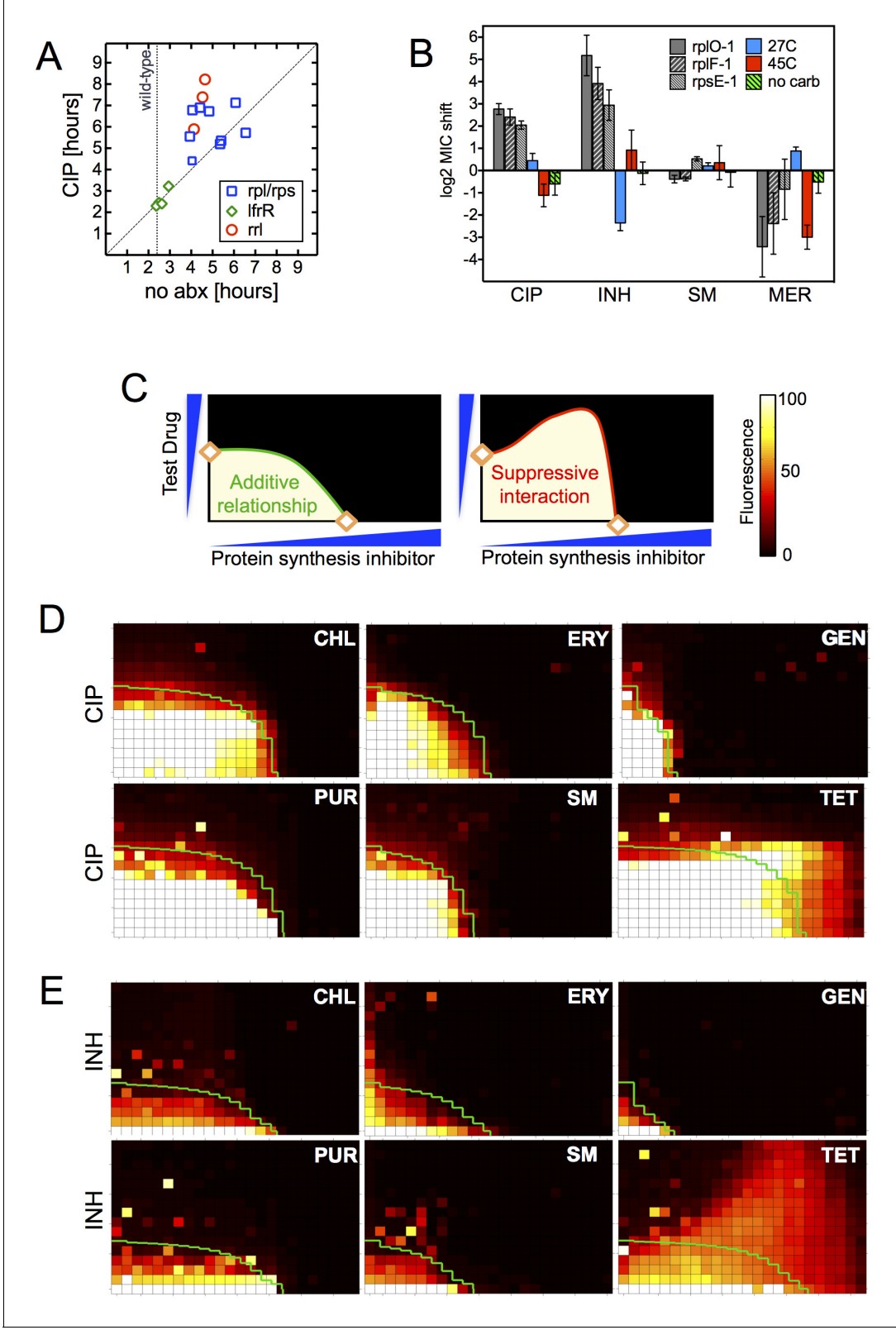

**Figure 3.** Resistance is not a simple consequence of slowed growth or general inhibition of translation. (**A**) Doubling times in antibiotic-free medium (x-axis) or 0.3 µg/ml CIP (y-axis), each dot represents an individual mutant. (**B**). Shifts in antibiotic sensitivity upon altering growth rate of wild type bacteria by temperature or carbon availability (dextrose/glucose-free medium). Error bars = 95% confidence intervals. See *Figure 3—figure supplement 1C* for growth rates. (**C**) Checkerboard assay description: 384-well plates contain increasing concentrations of protein synthesis inhibitors along the x-axis and

*Figure 3 continued on next page*

*Figure 3 continued*

a second test antibiotic on the y-axis. Growth of *M. smegmatis* mc$^2$155 (pUV3583cGFP) is measured by fluorescence. If the antibiotics are non-interacting then growth would follow the additive isobole in green (top box, see Supplementary Materials and methods). Antagonistic interactions extend growth beyond the isobole (bottom box). (D) Interaction of protein synthesis inhibitors chloramphenicol (CHL), erythromycin (ERY), gentamycin (GEN), puromycin (PUR) and streptomycin (SM) with CIP. (E) Same as (D) using INH. Tetracycline is unique in demonstrating antagonistic interactions with CIP and INH, phenocopying the ribosomal mutations.

The following figure supplement is available for figure 3:

**Figure supplement 1.** Growth rates of mutants and their relationship with resistance levels.

ribosomal mutants. Additionally, we looked to see whether the growth rates in our panel of mutants were correlated with the magnitude of the changes in antibiotic susceptibility (*Figure 3—figure supplement 1C*). Although growth was correlated with INH resistance ($R^2$ = 0.48) it was actually weakly inversely correlated with CIP and novobiocin (NOV) resistance ($R^2$ = 0.13, 0.14), showing that the higher IC50s of the mutants are cannot be accounted for solely by their slower growth.

Finally, in order to test whether impaired translation by mutant ribosomes could explain the shifts in antibiotic sensitivity, we inhibited ribosome function with six protein synthesis inhibitors: chloramphenicol, erythromycin, gentamycin, puromycin, streptomycin, and tetracycline. These drugs associate with the ribosome at different locations, interfering with various steps in translation (*Yonath, 2005*). If resistance were a general consequence of impaired translation, all would display an antagonistic relationship with CIP and INH. We performed 2D checkerboard experiments (*Figure 3C*), diluting a protein synthesis inhibitor horizontally and a test antibiotic (CIP or INH) vertically across 384-well plates, with the pattern of outgrowth indicating how the two antibiotics interact (*DePristo et al., 2007*; *Yeh et al., 2009*). Most protein synthesis inhibitors tested behaved additively with CIP or INH, thus ruling out general impairment of protein synthesis as the likely mechanism of resistance in these mutants. Only tetracycline significantly suppressed the activity of INH (*Figure 3D*) and CIP (*Figure 3E*). Among the inhibitors tested, tetracyclines act uniquely by inhibiting tRNA entry at the A-site (*Connell et al., 2003*), suggesting that this step in protein synthesis may be a common impairment among our ribosomal mutants, and that the consequences of such impairment are transduced to produce multidrug resistance.

## Genome-Scale studies illuminate the effects of ribosomal mutations on ribosome assembly

We used an unbiased approach to reveal changes in the baseline transcriptome and proteome of the ribosomal mutants that might account for the changes in antibiotic and stress sensitivity. We used RNAseq (*Pinto et al., 2011*) to characterize the baseline log-phase transcriptome of nine ribosomal mutants representing each of the major ribosomal loci, an *lfrR* mutant., and the parental mc$^2$155 (pUV3583cGFP) strain. Using DESeq2, we identified genes in each mutant that were differentially regulated relative to the parent. All of the ribosomal mutants showed abundant changes relative to the parent; on average, 1023 of 6718 transcripts (15%) were differentially expressed (range 471–2046), and these changes were highly overlapping as demonstrated by a core set of 227 genes that were significantly regulated in all nine ribosomal mutants (*Figure 4—figure supplement 2*). In contrast, of the 407 genes that were differentially regulated in the *lfrR* mutant, only 12 were among the core 227 shared by all of the ribosomal mutants. The similarity in behaviour among the different mutants as illustrated by their highly correlated expression profiles (*Figure 4A*, *Figure 4—figure supplement 1*) suggests a shared common mechanism for their multi-drug resistance phenotype.

In parallel, we used a comparative proteomics strategy, iTRAQ (isobaric tags for relative and absolute quantitation) (*Ross et al., 2004*), to identify proteomic differences between one representative ribosomal mutant, the rplO-1 mutant, and its wild type parent. The overlap between the changes in proteome of the rplO-1 mutant showed moderate correlation with the transcriptomic changes observed by RNAseq (Pearson = 0.55, p<0.001, *Figure 4B*). 64 genes were upregulated >2 fold in both datasets, while 41 were down >2 fold (*Figure 4—figure supplement 3*).

This iTRAQ data indicated a likely defect in ribosome assembly as a consequence of the rplO-1 lesion. iTRAQ revealed a reduction in the ratio of large ribosomal subunit proteins relative to small

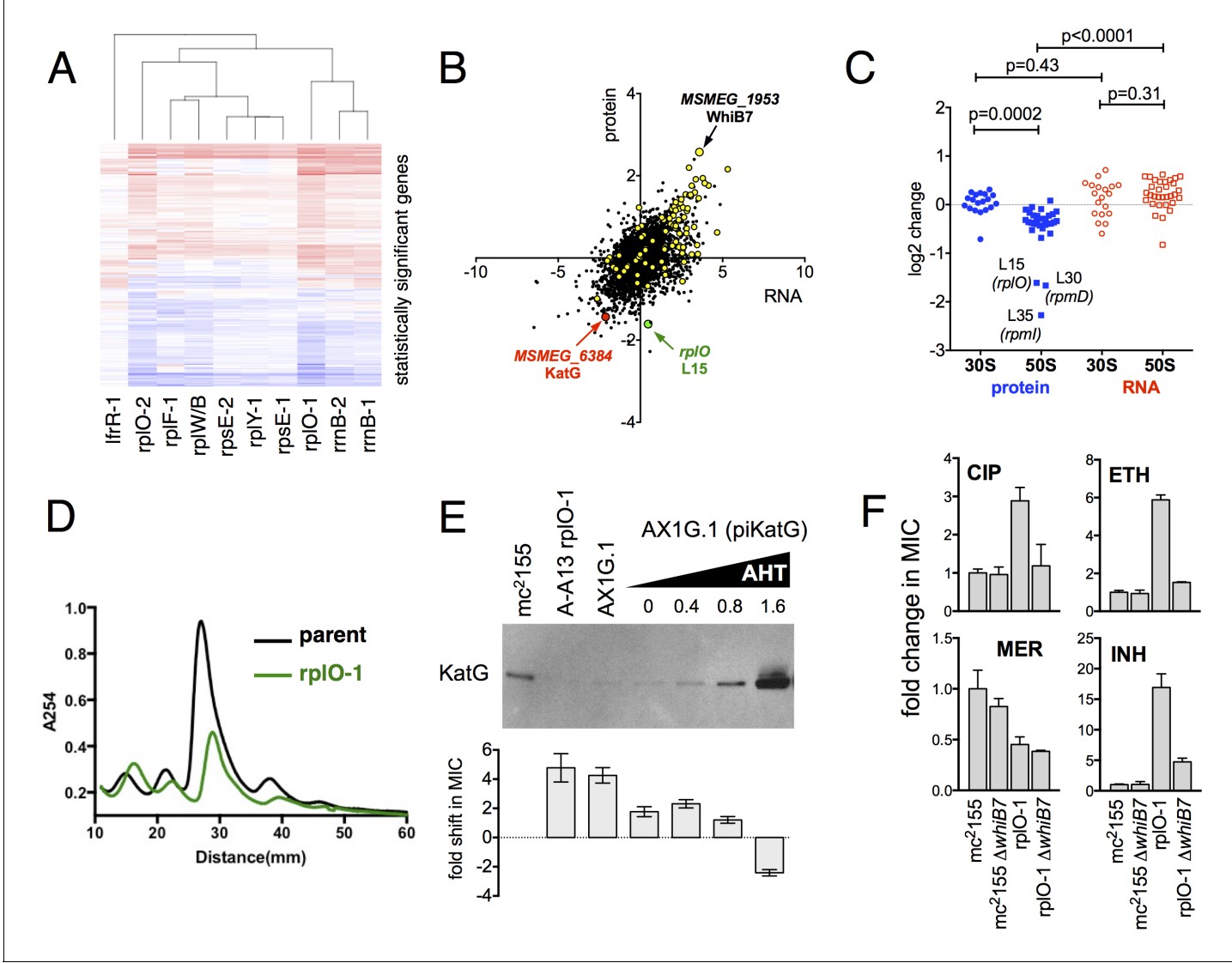

**Figure 4.** Alterations of the proteome and transcriptome implicate *katG* and the *whiB7* regulon in the altered antibiotic sensitivity of the ribosomal mutants; restoration of wild-type KatG levels restores INH sensitivity. (**A**) Hierarchical clustering of transcriptional alterations in representative ribosomal mutants, with lfrR-1 mutant included for comparison. For clustering, any genes whose expression is altered at least twofold and has an adjusted p value of <0.05 in any mutant is included. Source data are available as a Supplement. (**B**) Correlation between changes in RNA levels (x-axis) and protein levels (y-axis) in the rplO-1 mutant relative to wild type, plotted as log2 of fold change. Changes in both RNA and protein abundance are the averages of two replicate experiments. MSMEG_6384 (KatG) is highlighted in red and RplO (L15) is highlighted in green. Genes upregulated in MSMEG_6129 mutant (***Bowman and Ghosh, 2014***) (DOI: 10.1111/mmi.12448, supplementary file mmi12448-sup-0002-ts2.xls) are highlighted in yellow. See (**C**) Alterations in abundance of ribosomal proteins in the rplO-1 mutant relative to is parent strain as detected by iTRAQ (blue), and the corresponding changes in transcript abundance as determined by RNAseq (red). Student's T-test used for comparing datasets. (**D**) Comparison of ribosomal preparations separated on 10–40% sucrose density gradient, revealing impaired ribosome assembly in the rplO-1 mutant (green line). (**E**) Western blot of KatG protein levels in rplO-1 (A–A13), rplO-1 allelic exchange mutant AX1G.1, and rplO-1 AX1G.1 expressing KatG from an AHT-inducible plasmid at various AHT concentrations, shown above a plot of INH MIC under those conditions. As the expression of KatG is restored to wild-type levels, INH susceptibility is also restored. (**F**) Deletion of *whiB7* restores wild-type sensitivity to CIP and ETH but not INH or MER in an rplO-1 background. MICs (IC50) were calculated in PRISM using the ECAnything function to fit outgrowth across a 2-fold dilution series of each drug, and bars show mean ± SD of 2–4 biological replicates.

The following source data and figure supplements are available for figure 4:

**Source data 1.** Input for *Figure 4A*: DESeq2 output for all genes with padj <0.05 and twofold change in expression in any strain analyzed.

*Figure 4 continued on next page*

*Figure 4 continued*

**Figure supplement 1.** MA plots of the DESeq2 output relative to the parent for nine ribosomal mutants and *lfrR* mutant, highlighting *whiB7* (yellow dot) and *katG* (purple dots).
**Figure supplement 2.** Confirmation of transcriptional changes by PCR and impact of WhiB7 overexpression on antibiotic susceptibility.
**Figure supplement 3.** Sucrose density gradient profiles of ribosomes isolated from rplF-1, rpsE-1 and rplY-1 mutants, methods as described in
*Figure 4D*.

subunit proteins (*Figure 4C*) that was not reflected in the transcriptomic data, suggesting that the ribosomal mutations might have an effect on the stability or assembly of the large subunit, either decreasing its rate of translation or increasing its rate of degradation. To investigate whether ribosome assembly was perturbed across the various ribosomal mutants, we purified ribosomes from exponentially growing cultures of *rplO-1*, *rplF-1*, *rpsE-1*, *rplY-1*, and *rrlB1* mutants to evaluate the relative abundance of 30S, 50S and 70S subunits using sucrose gradient ultracentrifugation. Indeed, the profiles of the mutants indicate a marked increase in the individual 30S and 50S units and a corresponding decrease in fully assembled 70S subunits relative to the parent (*Figure 4D*, *Figure 4—figure supplement 3*).

To examine whether the ribosomal mutation and its impact on the ribosome could be inducing mistranslation as a cause of the multidrug resistance phenotype, we turned to the proteomic data obtained for the rplO-1 mutant compared to its parent. We found 7574 unique peptides with masses corresponding to mistranslated peptides (i.e. the mass corresponds to a known peptide with a single AA change). If mistranslation was occurring in the rplO-1 mutant, these mistranslated peptides should be more relatively abundant in the rplO-1 mutant, which should be reflected in the rplO-1/parent ratio for these peptides. However, the mean and median ratios seen in the mistranslated peptides are no different than the ratios seen across the larger set of 'normal' peptide; thus, mistranslated, antibiotic-insensitive targets are unlikely to contribute to resistance. In summary, the ribosomal mutations, while impairing 70S assembly, do not appear to lead to mistranslation; nevertheless, they cause an overall stress to the cell that results in a major reprogramming of the transcriptome and proteome of the bacteria, that ultimately lead to a multi-drug resistant phenotype.

## Genome-Scale studies suggest an overall transcriptional reprogramming contributing to multi-drug resistance

As the ribosomal mutations resulted in significant transcriptional reprogramming of the bacteria, we sought to look for individual clues that might account for the multi-drug resistance phenotype of the mutants. For example, the genome-scale data provided an immediate hypothesis to the nature of the observed INH resistance seen in the ribosomal mutants. Among the genes and gene products underexpressed in all strains by RNAseq and also in the iTRAQ dataset was MSMEG_6384, encoding the catalase KatG, which is required for converting the prodrug INH to its active form *Zhang et al. (1992)*. Relative to the parent strain. levels of *katG* transcript were reduced 7.1-fold in the rplO-1 mutant (adjusted p-value=$1.0 \times 10^{-54}$) and in all of the other ribosomal mutants (average of 3.9-fold across all of the mutants, minimum 2.5-fold, adjusted p-value=$1.1 \times 10^{-13}$), while iTRAQ showed a 4.2-fold reduction at the protein level in the rplO-1 mutant. The changes *katG* expression were confirmed by qRT-PCR (*Figure 4—figure supplement 1*) and Western blot analysis (*Figure 4E*). We hypothesized that among the dramatic overall transcriptome and resulting proteomic changes, the specific reduced KatG levels in the mutants could account for INH resistance. To test this, we introduced a plasmid (pIkatG) carrying an anhydrotetracycline (AHT) inducible copy of *katG* into an rplO-1 background. We then titrated the concentration of AHT to restore wild-type levels of KatG, measured by Western blot. Indeed, INH sensitivity was restored in the mutant background solely by restoring KatG levels back to wild-type levels, (*Figure 4E*), suggesting that INH resistance is due to the perturbations in KatG levels caused by the ribosomal mutations on the transcriptome and subsequently, on the proteome.

Similarly, we noted that among the global transcriptomic and proteomic changes occurring in the mutants, the induction of WhiB7, a transcription factor conserved throughout the actinomycetes that

positively regulates genes that contribute to innate antibiotic resistance, was among one of the strongest signals. As measured by iTRAQ, WhiB7 was one of the most highly induced proteins (6.0-fold, adjusted p-value=$2.5\times10^{-4}$) and by RNAseq, whiB7 mRNA was up 7.4-fold in rplO-1 relative to the parent strain (p adj.=$4.4\times10^{104}$) and in all of the other ribosomal mutants (average of 3.2-fold; minimum 1.7-fold, adjusted p-value=$2.1\times10^{-6}$). In M. smegmatis, whiB7 has been shown to be highly induced after exposure to specific classes of protein synthesis inhibitors, particularly tetracyclines and macrolides (Burian et al., 2012). We also observed high induction in the mutants of many genes known to be part of the whiB7 regulon, including several transporters that may contribute to antibiotic resistance by export (MSMEG_5187, a known antiporter of tetracycline (De Rossi et al., 1998), and MSMEG_5102). In addition to whiB7, there was strong correlation between the genes induced in our studies and genes previously shown to be upregulated in a strain lacking MSMEG_6129, a putative anti-sigma factor that is an indirect negative regulator of whiB7 ( 59) (Figure 4C, highlighted). The changes in whiB7 and katG expression were confirmed by qRT-PCR (Figure 4—figure supplement 1).

In order to investigate the functional contribution of the overexpression of whiB7 to the observed antibiotic resistances, we first deleted the whiB7 gene in both a wild-type and an rplO-1 background. In the rplO-1 background, deletion of whiB7 restored sensitivity to a CIP, ethionamide (ETH) and cycloserine (CYC) (Figure 4C), indicating a critical role for WhiB7 in the broad-spectrum resistance observed in the rplO-1 mutant. Deletion of whiB7 however, did not fully restore INH sensitivity to rplO-1, nor did it alleviate the increased sensitivity to meropenem. We then constitutively expressed WhiB7 at a high-level from a plasmid in a wild-type background and found that it alone did not confer resistance to CIP, INH, ETH, or MER (Figure 4—figure supplement 2). Thus, WhiB7 induction does play a role in the multi-resistance phenotypes; however, it alone is insufficient to account for the entire resistance profiles of the ribosomal mutants. It must act in concert with other pathways to exert its effect on CIP, ETH, MER and INH resistance, with the example of INH resistance resulting via additional alterations in katG expression.

## Ribosomal mutations promote the evolution of high level resistance

We next turned to understand whether ribosomal mutations can contribute to the evolution of high-level multidrug resistance. We first investigated if ribosomal mutants can emerge directly from a multidrug environment, We performed selections in CIP and INH administered in combination (Figure 5A). Mutants arose during double selection at a 3600-fold higher frequency ($5.5 \times 10^{-7}$) than the product of their frequencies in CIP ($1.5 \times 10^{-5}$) and INH ($1.0 \times 10^{-5}$) alone, suggesting that the mutants that arose in the presence of both CIP and INH are not the product of two completely uncoupled mechanisms. When we PCR amplified and sequenced a fragment including the rplF, rpsE and rplO genes from three mutants that arose under double selection, we identified the same frameshift mutations in rplO in two of these that we had previously observed in rplO-2 (77ΔA) and rplO-3 (74-77ΔGTGA).

Ribosomal mutations should also promote the acquisition of high-level antibiotic-specific resistance (e.g., gyrA or lfrR) due to the larger population of surviving cells from which resistant mutants can emerge (Figure 5B). In support of this, we found that upon selection at 3.5 µg/ml CIP, a bactericidal concentration for both wild-type cells and the ribosomal mutants (Figure 5B), strains with mutations in rplO or rplF yielded 10–100 fold more high-level CIP resistant mutants per input cell than wild-type cells (Figure 5C). When we sequenced the gyrA and lfrR loci in these mutants, secondary mutations in both of these genes were identified (Table 2, strains rplO-lfrR-1 through six and rplO-gyrA-1).

While the ribosomal mutations result in a fitness cost in antibiotic-free conditions (Figure 3A), acquisition of target- or efflux-associated mutations renders the ribosomal mutations dispensable. Subsequent acquisition of compensatory changes could restore fitness, thereby relegating the ribosomal mutations to a role as an evolutionary 'bridge' across periods of exposure to antibiotic or other stress. To test for this possibility, we first sought to determine if compensatory changes that restore fitness could occur in the absence of antibiotic selection. We serially passaged three parallel cultures of the rplO-1 mutant in antibiotic-free medium. By passage 12, cultures demonstrated wild-type growth rates (Figure 5D) as well as wild-type sensitivities to CIP and INH (Figure 5E). Evolved strains in all three cultures had acquired an additional insertion or deletion at the rplO locus (Figure 5F, Table 2), restoring the wild-type reading frame but leaving short deletions or coding

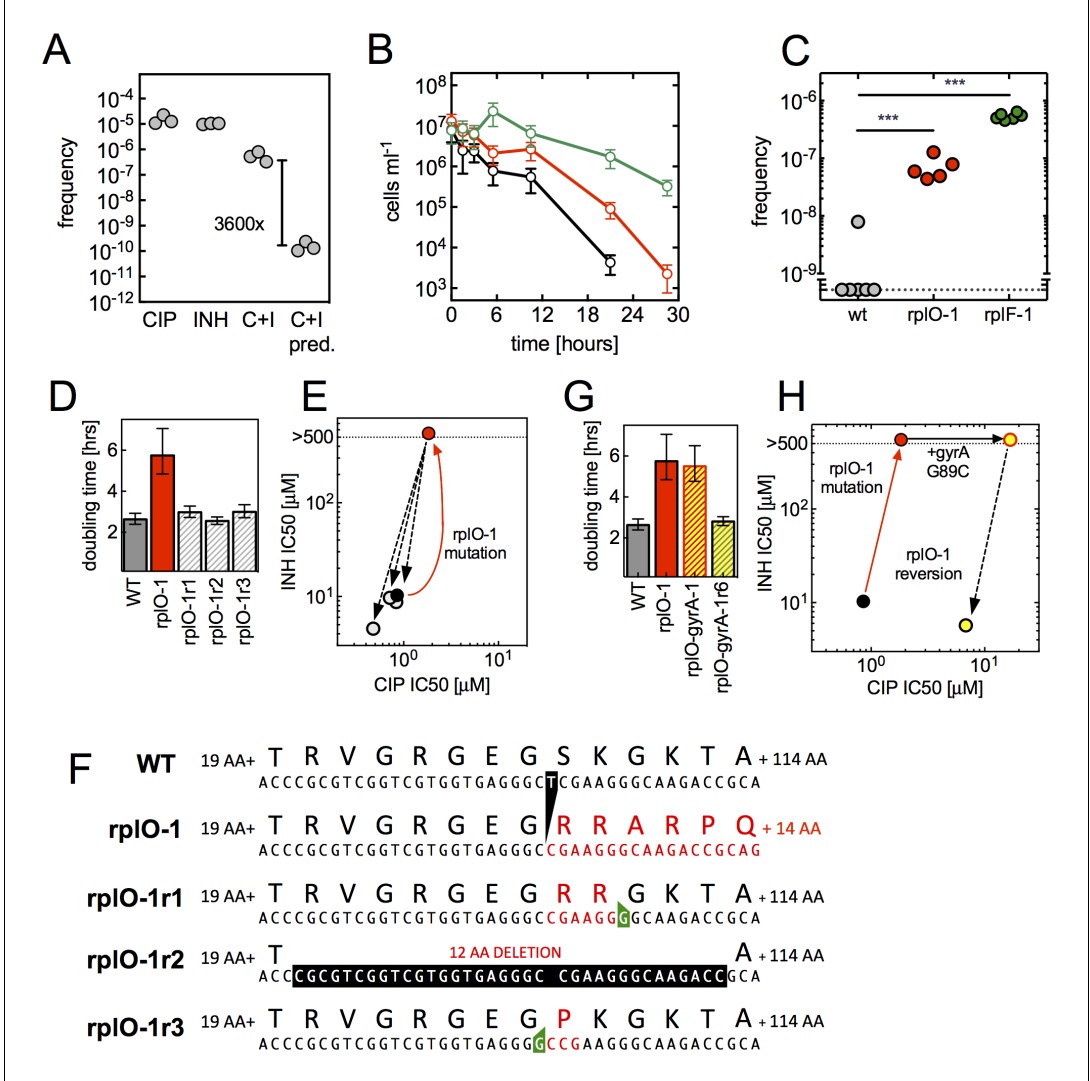

**Figure 5.** An evolutionary cycle for the acquisition of high-level resistant mutants without fitness cost. (A) Ribosomal mutations arise during simultaneous selection in two antibiotics. Frequency (per input cell) of resistant mutants from three replicate cultures (individual dots) in CIP (0.3 µg/ml, C), INH (25 µg/ml, I) or CIP and INH in combination (0.3 µg/ml / 25 µg/ml, C+I). C+I predicted = the product of the measured frequencies for each drug individually. (B) Prolonged survival of rplO-1 (red), or rplF-1 (green) relative to wild-type (black) bacteria exposed to 3.5 µg/ml CIP in 384 well plates. (C) Increased frequency of emergence of CIP mutants resistant to >3.5 µg/ml CIP derived from each of the genetic backgrounds in (B), ** = p<0.05, student's T-test. (D) The growth defect of an *rplO-1* mutant (red bar) is reversed after serial passage of 3 independent cultures (hatched bars) to levels similar to wild-type cells (gray bar) (E) Restoration of wild-type CIP and INH sensitivities after serial passage (shading as in D). Dots represent strains shown in (D), with the mutational trajectory indicated by arrows. The x and y axes show the level of susceptibility to CIP and INH. (F) Deletion in *rplO* (black) leads to a shifted rplO-1 reading frame (red). Compensatory insertions (green) or deletions (black) in rplO-1r1-3 restore the original reading frame (blue). (G) Growth defect of the *rplO-1* mutation (red bar) is maintained after acquisition of a *gyrA* mutation (rplO-1-gyrA, yellow and red hatched bar). Continued passage of rplO-1-gyrA in CIP restores wild-type growth rates (solid yellow bar) (H) Loss of INH resistance after continued selection in CIP Dots represent strains shown in (G), with the mutational trajectory again indicated by arrows. The x and y axes show the level of susceptibility to CIP and INH. (shading as in G).

changes in the region between the two sequential mutations. We then demonstrated that reversion to wild-type growth rates can also occur readily in a *rplO gyrA* double mutant (rplO-1 gyrA-1, *Table 2*) during continued passage at 3.5 µg/ml CIP (*Figure 5G–H*, *Table 2*); INH resistance is lost as a consequence of a compensatory one base insertion that restores the original reading frame but alters one codon in *rplO*. Thus, these secondarily acquired mutations within *rplO* that restore the

**Table 2.** Evolved strains with additional mutations.

| Name | Parent strain | Phenotype | Second mutation | Consequence |
|---|---|---|---|---|
| rplO-lfrR-1 | rplO-1 (A-A13) | CIP MIC > 3.5 µg/ml | lfrR 427+C | LfrR frameshift |
| rplO-lfrR-2 | rplO-1 (A-A13) | CIP MIC > 3.5 µg/ml | lfrR 440+C | LfrR frameshift |
| rplO-lfrR-3 | rplO-1 (A-A13) | CIP MIC > 3.5 µg/ml | lfrR T128G | LfrR V43G |
| rplO-lfrR-4 | rplO-1 (A-A13) | CIP MIC > 3.5 µg/ml | lfrR 172+ACC | LfrR 58+H |
| rplO-lfrR-5 | rplO-1 (A-A13) | CIP MIC > 3.5 µg/ml | lfrR 440+C | LfrR frameshift |
| rplO-lfrR-6 | rplO-1 (A-A13) | CIP MIC > 3.5 µg/ml | lfrR 118+G | LfrR frameshift |
| rplO-gyrA-1 | rplO-1 (A-A13) | CIP MIC > 3.5 µg/ml | gyrA G265C | GyrA G89A |
| rplO-1r1 | rplO-1 (A-A13) | wt growth rate | rplO 89+G | L16 SK28-29RR |
| rplO-1r2 | rplO-1 (A-A13) | wt growth rate | rplO 61-96Δ* | L16 ΔRVGRGEGRRGKT |
| rplO-1r3 | rplO-1 (A-A13) | wt growth rate | rplO 80+G | L16 S28P |
| rplO-gyrA-1r6 | rplO-gyrA-1 | wt growth rate | rplO 89+G | L16 SK28-29RR |

*=CGCGTCGGTCGTGGTGAGGGCCGAAGGGCAAGACC.

reading frame provide a facile, straightforward mechanism for restoring bacterial fitness after a higher level resistance lesion, *i.e., gyrA* mutation, has been acquired.

Finally, one well in our initial multi-well selection experiments illustrates how rapidly the transition from low to high-level resistance can occur. Fluorescence was detected in well A-G20 after 5 days of CIP exposure, a timing consistent with the slow emergence of ribosomal mutants in this assay. The culture expanded from A-G20 was also resistant to INH, NOV, ETH and CYC, consistent with a ribosomal mutation. However, after further expansion for genomic DNA preparation, WGS revealed SNPs in both *lfrR* (T452C) and *rrlB* (C904A). Additionally, MIC assays showed high level resistance to CIP, consistent with the presence of an *lfrR* efflux mutation. To better understand how these two SNPs came to coexist in the A-G20-derived culture, we colony-purified 23 clones from the frozen stock originally expanded from well A-G20 and used PCR to characterize the *rrlB* and *lfrR* loci (*Table 3*). While the majority (19/23 clones) had only the *rrlB* C904A mutation, three had both the *rrlB* C904A and *lfrR* T452C mutation. Surprisingly, the final clone had the *rrlB* C904A mutation paired with a different *lfrR* allele, 63(+C). Thus, the *rrlB* C904A mutation likely arose first, enabling slow replication in the presence of CIP and accounting for the delayed emergence of well A-G20 on day 5. Continued selective pressure during the remainder of the multiwell assay and during subsequent expansion in 0.3 µg/ml CIP enabled the eventual emergence of *lfrR* mutants from the *rrlB* C904A background. These second mutations, providing high level CIP resistance, would increase fitness in the presence of CIP; as seen in *Figure 3A*, the growth rate of *rrlB* mutants is slower in the presence of 0.3 µg/ml CIP than in antibiotic-free medium. Thus, a single cycle of selection and expansion was sufficient to demonstrate the evolutionary value of ribosomal mutations to the development of high level resistance.

## Discussion

With increasing antibiotic resistance, there is increased urgency to understand how on the molecular and cellular level, bacteria evolve to acquire resistance. Despite strategies such as combination therapy, resistance is nevertheless arising in complex, multidrug environments. In this work we have identified a novel class of mutations in ribosomal subunits and rRNA that surprisingly confers resistance to several mechanistically diverse antibiotics. These alterations in the ribosome do not result in mistranslation, yet cause misassembly of the ribosome, leading to a dramatic shift in the transcriptional program of the bacterium by that imparts resistance to multiple antibiotics with different mechanisms of action. Within the large-scale changes are numerous, discrete changes in individual genes that result in resistance to specific antibiotics. For example, changes in *katG* expression confer INH resistance while induction of *whiB7* plays a role in altering susceptibility to CIP, ethionamide, and cycloserine. Single mutations in the ribosome allow a bacterium, when simultaneously exposed

**Table 3.** Genotypes of individual colonies from freezer stock derived from well A-G20. Colony 15 failed to grow.

| Colony | *rrlB* genotype | *lfrR* genotype |
| --- | --- | --- |
| A-G20-1 | *rrlB* C904A | wt |
| A-G20-2 | *rrlB* C904A | wt |
| A-G20-3 | *rrlB* C904A | wt |
| A-G20-4 | *rrlB* C904A | wt |
| A-G20-5 | *rrlB* C904A | wt |
| A-G20-6 | *rrlB* C904A | wt |
| A-G20-7 | *rrlB* C904A | wt |
| A-G20-8 | *rrlB* C904A | wt |
| A-G20-9 | *rrlB* C904A | wt |
| A-G20-10 | *rrlB* C904A | wt |
| A-G20-11 | *rrlB* C904A | wt |
| A-G20-12 | *rrlB* C904A | wt |
| A-G20-13 | *rrlB* C904A | wt |
| A-G20-14 | *rrlB* C904A | *lfrR* T452C |
| A-G20-15 | ND | ND |
| A-G20-16 | *rrlB* C904A | wt |
| A-G20-17 | *rrlB* C904A | wt |
| A-G20-18 | *rrlB* C904A | *lfrR* T452C |
| A-G20-19 | *rrlB* C904A | wt |
| A-G20-20 | *rrlB* C904A | wt |
| A-G20-21 | *rrlB* C904A | wt |
| A-G20-22 | *rrlB* C904A | *lfrR* 63+C |
| A-G20-23 | *rrlB* C904A | wt |
| A-G20-24 | *rrlB* C904A | *lfrR* T452C |

to lethal concentrations of multiple antibiotics (CIP and INH), to survive, grow, and ultimately acquire additional mutations that confer high-level resistance. These mutations are thus beneficial in multi-drug environments.

Even the low to moderate levels of resistance in the ribosomal mutants are sufficient to confer a survival advantage over wild-type cells in the presence of antibiotic(s). Whether the mutations allow them to survive in general or simply to prolong their time to death, this extended survival provides the ribosomal mutants an increased opportunity, relative to wild-type bacteria, to acquire a second mutation to become high-level resistant. Thus the presence of a small population of ribosomal mutants could contribute to the survival of the larger population, especially in antibiotic concentrations fluctuating near or above the MIC (as might be seen during treatment of a bacterial infection), and facilitate fixation of the second high-level resistance conferring mutation in the population.

These mutations in the ribosome do carry a fitness cost in the absence of antibiotic. However, the nature of these particular mutations are remarkable because locus-specific reversions can easily occur to restore fitness upon passage in conditions where these mutations no longer confer an advantage. Such conditions include an antibiotic-free environment or the continued presence of an antibiotic but after the acquisition of a second, higher-level resistance-conferring mutation that renders the ribosomal mutation dispensable. The nature of the identified lesions (frameshifts in protein coding genes and substitutions in *rrlB*) and their respective reversions (restoration of the original reading frame) illustrate the straightforward means for regaining fitness via additional insertions/deletions or via gene conversion. This is particularly facile in the case of the 23S rRNA mutations

because like many bacteria, *M. smegmatis* has multiple rRNA operons (two). Resistance-conferring *rrlB* mutations were acquired only in one copy of the rRNA, thereby facilitating, RecA-mediated gene conversion from the other, wild-type copy (*Prammananan et al., 1999*). Similar to the way reversions contribute to the evolution of cefotaxime resistant ß-lactamases by removing mutations that alone increase fitness but are deleterious in combination with certain other fitness-enhancing mutations (*DePristo et al., 2007*), the ability to revert in this manner greatly expands the number of possible trajectories to resistance. Thus, these particular mutations make accessible an efficient evolutionary path to multidrug resistance via a temporary state of impaired fitness that is ultimately reversed to restore wild-type fitness.

Prokaryotic ribosomes are complex molecular machines in which we now show that alterations can confer resistance to a range of mechanistically diverse antibiotics. The two ribosomal subunits together contain over 50 proteins, the functions of which have not yet been fully elucidated. The majority of these proteins are expected to be essential, although recent studies suggest in *E. coli* and *B. subtilis* that as many as 22 may be conditionally dispensable (*Shoji et al., 2011*; *Akanuma et al., 2012*). Among these non-essential proteins are two identified in this study that can play a role in multidrug resistance, namely L15 (RplO) and L25 (RplY). The other ribosomal proteins identified in our study, L6 (RplF) and S5 (RpsE), are essential in these organisms. Transposon-based studies of gene essentiality in the related mycobacterium *M. tuberculosis* (*Sassetti et al., 2003*; *Zhang et al., 2012*) show that *rplF* and *rplE* are also essential in *M. tuberculosis*, while *rplY* was non-essential and the essentiality of *rplO* was indeterminate. The mutations we found in *rplF* and *rpsE* occur near the 3' end of the open reading frames, leaving the majority of the corresponding proteins, L6 and S5 unaltered.

Our drug interaction experiments using a range of protein synthesis inhibitors potentially link these mutations with the action of tetracycline. Although tetracycline binds to the ribosome at several locations (*Day, 1966*), its primary binding site is located near the A site, where it interacts with 16S RNA in the 30S subunit (*Brodersen et al., 2000*). At this site, tetracycline inhibits protein synthesis by interfering with tRNA entry (*Wurmbach and Nierhaus, 1983*; *Maxwell, 1967*). Thus, the ribosomal mutations we have identified may have subtle effects in translation initiation that are similar to the effects of low concentrations of tetracycline. These are then sensed by the cell, leading to a large shift in the overall transcriptome, including the specific induction of *whiB7* and other specific transcriptional alterations that affect antibiotic sensitivity to other antibiotics. As a corollary, this work raises the question of whether tetracyclines may more generally interact with antibiotics negatively to impact sensitivity.

While traditional culture conditions in the laboratory select for strains with ribosomal properties that maximize growth rates, the varied environments encountered by microbes in nature may select for other ribosomal properties; the slower growth rates of natural isolates of *E. coli* were shown to be well-correlated with the kinetic properties of their ribosomes, and extended axenic culture of these isolates led to the evolution of culture-adapted strains with growth rates similar to those seen in laboratory strains (*Mikkola and Kurland, 1991*). The existence of natural bacterial isolates with submaximal ribosome kinetics implies that under certain conditions, suboptimal or altered ribosome function may in fact be beneficial. By linking alterations in ribosome assembly with broad transcriptional reprogramming that enhances survival in fluctuating environments, *M. smegmatis* and other actinomycetes may benefit from ribosomal variants, even at the cost of decreased maximal growth rate. Via genetically accessible paths to reversion, these populations can serve as stepping-stones to higher levels of drug resistance without any irreversible cost in fitness. Because environmental or pathogenic bacteria face extended or fluctuating periods of stress, the generation of minority populations that are resistant to a broad range of antibiotic and other stresses could serve as an evolutionary bridge to the eventual emergence of fit, antibiotic resistant strains.

## Materials and methods

### Strains and media

*Mycobacterium smegmatis* mc$^2$155 was grown at 37°C in Middlebrook 7H9 medium (M7H9) supplemented with ADS (Albumin-Dextrose-Saline: Bovine Serum Albumin (Fraction V) 5 g/liter, dextrose 2 g/l, sodium chloride 0.85 g/l final concentration) and Tween 80 (0.05%). LB agar was employed for

growth on solid medium. To generate a fluorescent strain, mc$^2$155 was transformed via electroporation with pUV3583cGFP, a derivative of pUV15tetORm (*Ehrt et al., 2005*) in which the tetracycline regulated promoter P$_{myc1}$ was replaced by a 525 bp fragment of *M. tuberculosis* gDNA carrying the promoter for the highly-expressed *carD* gene. *Mycobacterium smegmatis* mc$^2$155 was obtained from Keith Derbyshire at the University of Albany.

## Selection of resistant mutants

Bacteria were seeded at a density of 10 organisms/well into 15 µl of M7H9 in clear-bottom plates using a FACS Aria II SORP flow cytometer (Becton Dickinson) and allowed to expand overnight prior to addition of ciprofloxacin at a final concentration of 0.3 µg/ml (1.2X the measured MIC). Plates were sealed with foil and incubated at 37°C without shaking. Growth of resistant mutants was assessed by measuring fluorescence accumulation in each well (bottom read, excitation 485 nm, emission 538 nm, cutoff 530) on a roughly daily basis. Plates were monitored for a total of 12 days. On day 12, fluorescent wells were harvested and expanded to 3 ml in the continued presence of 0.3 µg/ml CIP. An aliquot was frozen at −80°C and the remaining contents were again expanded to 25 ml in 0.3 µg/ml CIP for the purpose of gDNA preparation.

## Bacterial viability measurements

The viability of cells during initial selection was measured using a most probable number (MPN) assay (*Cochran, 1950*). Cells were aliquotted into 4–8 replicate wells in a Costar 96-well black clear bottom plates and then serially diluted ten fold. Plates were grown for approximately 10 days to allow wells in which only single cells were transferred to become turbid. Plates were then read in M5 SpectraMax plate reader (excitation 485, emission 538, cutoff 530) and MPN subsequently calculated. The 95% confidence interval of the MPN assay is a function of the pattern of observed outgrowth. For heat and SDS stress experiments, viability was measures by serial dilution in 4-fold steps in growth medium followed by counting of colonies after spotting 5 µl using a dissecting microscope.

## Antibiotic sensitivities

To ascertain antibiotic sensitivity 1.41-fold ($\sqrt{2}$) dilutions of antimicrobials were prepared and seeded with cells at optical densities of 0.02, then grown for 2–3 days. Cell numbers were measured with GFP when available or optical density otherwise. The concentration inhibiting outgrowth to 50% of the no antibiotic control, or the *IC50*, was determined by fitting a Hill function to the outgrowth curve of the following form:

$$y_M = b + \frac{(m-b)}{1 + (x_M/IC50)^n}$$

where three additional parameters are fit: the maximum outgrowth *m*, the baseline outgrowth *b*, and the Hill coefficient *n*. Fit parameters are constrained such that *m* > *b* > 0, *n* > 0, and *IC50* > 0. Fitting was performed on log-transformed data, and then transformed back to a linear axis, preventing least-squares fitting from being dominated by expression values near the maximum.

## Sequencing of mutants

Genomic DNA was extracted using the cetyltrimethylammonium bromide (CTAB)-lysozyme protocol described (*van Soolingen et al., 1991*). For determination of *gyrA* and *lfrR* sequences, these loci were amplified using the primers lfrR F, lfrR R, gyrA QRDR F1 and gyrA QRDR R1and sequenced at the MIT Biopolymers Laboratory. Whole-genome sequencing of an initial set of 8 strains, including A-M17, A-L8, A-A13, A-H2, B-03, B-I24, B-A9 and B-P7, was performed at Texas A and M University using an Illumina GenomeAnalyzer IIx, as described in *Ioerger et al. (2010)*. Fragment libraries were constructed using a genomic DNA sample preparation kit from Illumina, Inc. Sequencing data was collected in paired-end mode (from both ends of each fragment) with a read length of 36 bp. Reads were aligned to the genome of *M. smegmatis* mc$^2$155 as a reference sequence (Genbank accession number NC_008596.1). Insertions and deletions were determined using a local contig-building algorithm, and then single-nucleotide polymorphisms were identified from a re-alignment of the reads. The depth of coverage ranged between 23x to 42x (mean number of reads covering each site in the

genome). Strains A-A15, A-A24, A-G16, A-G20, A-K20, A-K24, A-L8, A-M4, B-O11, B-O18, B-P7, B-P17 were subsequently sequenced at the Broad Institute, also using Illumina materials for library preparation materials and Illumina instrumentation for data generation.

Confirmation of ribosomal lesions by PCR and sequencing was done using the following primers (see *Supplementary file 1* for sequences): *rplO*: rplO F1 and rplO R1; *rplF*: rplF F1 and rplF R1; *rplY*: ctc F1 and ctc R1; *rpsE*: rpsE1 and rpsER1, rplW/B: rplW/B seqF1 and rplW/B seqR1. Because *M. smegmatis* has two ribosomal RNA operons, *rrnA* and *rrnB*, whole genome sequencing data reflected the presence of both wild-type and mutant alleles, but was insufficient to determine which of the two operons contained the mutated *rrl*. To address this, a conserved primer within the operon, rrl-F1, was paired with either of two unique primers from the region downstream of the operon to amplify the rrlB gene in allele-specific manner (rrlA-R1 or rrlB-R1.) The primers rrl-F2, rrl-F3, rrl-F4, rrl-F5, rrnb_s1, rrnb_s2, rrnb_s3 were used to sequence these products.

## Generation of allelic exchange mutants

For ribosomal mutants, chromosomal regions from isolate A-A13 (rplO-1) and A-K20 (rpsE-1) were amplified using primers MS_rplO_AX F, MS_rplO_AX R, Ms_rpsE_AX F, and Ms_rpsE_AX R. PCR products were digested with *PacI* and *AscI* and cloned into pJG1100, a suicide vector carrying *aph* (kanamycin resistance), *hyg* (hygromycin resistance), and *sacB* (sucrose sensitivity). For whiB7 null mutants, flanking regions were amplified with primers Ms_whiB7KO_USr2_Pac1, Ms_whiB7KO_US-f2_Asc1, Ms_whiB7KO_DSf1_Not1, Ms_whiB7KO_DSr1_Pme1 and cloned into pJG1100 in two steps after PacI-AscI and PmeI DraI digestion to yield a vector in which *whiB7* is replaced by a hygromycin resistance cassette. After sequence confirmation, plasmids were electroporated into *M. smegmatis* mc$^2$155, and integrants (single recombinants) were selected on LB containing 50 µg/ml hygromycin B and 12.5 µg/ml kanamycin. Integrants were picked into 7H9 medium and grown overnight prior to plating on LB containing 2.5% sucrose. Colonies were picked into 96 well plates and grown until visibly turbid in 7H9 prior to spotting on LB containing 50 µg/ml hygromycin B (for ribosomal mutants) or kanamycin (for *whiB7* null mutant). Hygromycin sensitive colonies were screened for the presence of the mutant allele by PCR amplification of the relevant ribosomal gene. Complementation of the rplO-1 allelic exchange mutant was carried out by transforming that strain with pUV15tet-rplO, a pUV15tetORm derivatave in which *rplO* expression is controlled via a tetracycline-regulated promoter. The *M. smegmatis rplO* open reading frame was PCR-amplified using the primers MS_rplO_5'_wt and MS_rplO_orf_3' and this fragment was digested with PacI and EcoRV and cloned into similarly-cut pUV15tetORm. Transformants were selected on hygromycin and production of functional L15 protein was induced using 50 ng/ml anhydrotetracycline (AHT).

## Measurement of replication rates

We measured the growth rates of our ribosomal mutants in microtiter plates, the environment in which the mutations were generated, using a serial dilution method. Aliquots of each strain were diluted in a two-fold series across a 384-well plate and the accumulation of fluorescence in each well was assessed automatically approximately every hour. We then determined the time at which each dilution crossed a threshold, roughly 3-fold above the auto-fluorescence background. The delay in crossing this threshold between adjacent twofold dilutions reflects the doubling time in this format. To confirm that allelic exchange mutants showed similar impairments in growth to the original mutants, growth rates were assessed by periodic sampling of exponentially growing 30 ml cultures.

## Analysis of ribosome populations

100 ml of culture at OD600 = 0.4–0.6 was treated with 50 µg/ml of chloramphenicol (CAM) to freeze ribosome activity. Cells were pelleted and resuspended in buffer TM (20 mM Tris pH7.5, 15 mM MgCl$_2$, 1 mg/ml PMSF, 50 µg/ml CAM). Cells were lysed by bead beating. 20 µl of 10% deoxycholate was added to lysates, which were clarified at 15,000 rpm for 15 min. 5 mg of cleared lysate (500 µl of 10 mg/ml cleared lysate) was layered onto a 10–40% sucrose gradient in Buffer E (10 mM Tris, 10 mM MgCl$_2$, 100 mM NH$_4$Cl, 3 mM $\beta$-mercaptoethanol, 50 ug/ml CAM). The sucrose gradient was spun in a Beckman ultracentrifuge at 35,000 rpm (150,000xg) for 2.5 hr at 4°C. The gradient was fractionated using a gradient fractionator (BioComp Instruments, Inc., NB, Canada).

## Checkerboard experiments

The ability of small-molecule protein synthesis inhibitors to phenocopy the effects of ribosomal mutations was assayed in 384 well microtiter plates, using *M. smegmatis* (pUV3583cGFP), with fluorescence as a readout. Along the short axis of two plates (31 steps, with no antibiotic in the final row of plate 2) a readout antibiotic (or SDS) was diluted such that each successive wall was 0.8X as concentrated as the next. Likewise, along the long axis (23 steps, with no antibiotic in column 24), a protein synthesis inhibitor was similarly diluted. Readout antibiotics included CIP, INH, MER, and streptomycin (SM), and modulator antibiotics including tetracycline (TET), erythromycin (ERY), SM, puromycin (PUR) and gentamycin (GM).

A common metric used to define drug-drug interactions is Loewe additivity, a concept that stems from the notion that a given drug should neither synergise nor antagonize when placed in combination with itself. While intuitive, competing notions of additivity, such as Bliss independence, do not follow this intuitive idea because the response of bacteria to drugs can be highly nonlinear.

We define the additive curve between two drugs, *DrugX* and *DrugY*, as a linear function of the form:

$$c_{DrugY} = m \cdot c_{DrugX} + b$$

where *m* is the slope, *b* is the curve's intercept with the y-axis, and $[c_{DrugX}]$ and $[c_{DrugY}]$ are concentrations of the two drugs. We constrain this curve by demanding that when DrugX is absent (i.e., $[c_{DrugX}] = 0$) the concentration of *DrugY* is equal to its *IC50* concentration ($IC50_{DrugY}$), and vice versa. Using these as boundary conditions we solve for the intercept, *b*:

$$b = IC50_{DrugY}$$

and the slope, *m*:

$$m = -\frac{IC50_{DrugY}}{IC50_{DrugX}}$$

so that finally

$$[c_{DrugY}] = -\frac{IC50_{DrugY}}{IC50_{DrugX}} \cdot [c_{DrugX}] + IC50_{DrugY}$$

In the limiting case where *DrugX* and *DrugY* are the same chemical compound (so a checkerboard experiment is performed with the same drug diluted in both axes) then $IC50_{DrugX} \equiv IC50_{DrugY}$. The equation above then reduces to the simple form:

$$[c_{DrugY}] + [c_{DrugX}] = IC50_{DrugY}$$

which satisfies Loewe's criterion where a single drug has no synergistic or antagonistic effect on itself. In our figures the additive curve appears as an arc, not a straight line, because the both axes are displayed in a log scale.

## Transcriptional analyses

For total RNA preparation for RNA seq, 5 ml replicate cultures *of M. smegmatis* were grown to midlog phase and pelleted at 3500x g. Pellets were resuspended in 0.9 ml Trizol and frozen at −80C. After thawing, 0.4 ml of zirconia-silica beads were added and cells were lysed by bead-beating. Bead-beaten tubes were centrifuged at 12,000x*g* for 15 min. 0.4 mL of aqueous phase was added to equal volume of 100% EtOH, vortexed, and loaded onto Zymo DirectZol columns and RNA was purified according to the manufacturer's protocol. Illumina cDNA libraries were generated using a modified version of the RNAtag-seq protocol as described (*Shishkin et al., 2015*). Briefly, 500 ng of total RNA was fragmented, depleted of genomic DNA, and dephosphorylated prior to its ligation to DNA adapter carrying 5'-AN8-3' barcodes with a 5' phosphate and a 3' blocking group. Barcoded RNAs were pooled and depleted of rRNA using the RiboZero rRNA depletion kit (Epicentre). These pools of barcoded RNAs were converted to Illumina cDNA libraries in three main steps: (i) reverse transcription of the RNA using a primer designed to the constant region of the barcoded adaptor; (ii) addition of a second adapter on the 3' end of the cDNA during reverse transcription using

SmartScribe RT (Clontech) as described (*Trombetta et al., 2014*); (iii) PCR amplification using primers that target the constant regions of the 3' and 5' ligated adaptors and contain the full sequence of the Illumina sequencing adaptors. cDNA libraries were sequenced on Illumina HiSeq 2500. For the analysis of RNAtag-Seq data, reads from each sample in the pool were identified based on their associated barcode using custom scripts, and up to one mismatch in the barcode was allowed with the caveat that it did not enable assignment to more than one barcode. Barcode sequences were removed from reads, and the reads from each sample were aligned to genes in their cognate strain using BWA (*Li and Durbin, 2010*). Differential expression analysis was conducted with raw reads counts per gene using DESeq2 (*Love et al., 2014*).

## Proteomic analyses

Duplicate 30 ml cultures of mc$^2$155 and rplO-1 were harvested in mid-log phase by pelleting at 4000xg and washed twice in cold PBS + 0.05% Tween 80, followed by two additional washes in cold PBS without Tween 80. Pellets were resuspended in 750 µl PBS and beaten for 3 × 1 min with zirconia –silica beads to lyse the cells. Lysates were centrifuged for 20 min at 4°C at 16,000xg to separate soluble proteins from beads and cell debris/unlysed cells. These supernatants were removed and transferred to new 1.5 ml tubes, The silica pellets were rinsed in an additional 150 µl of PBS, vortexed 15 s. The supernatants and rinsed beads were centrifuged an additional 20 min at 16,000xg (a small pellet was seen in the supernatant), and then approx. 700 µl of cleared supernatant was transferred to a fresh tube. The supernatants from the rinsed beads were transferred to new tubes and centrifuged an additional 20 min at 16,000xg. Again a small pellet was seen in the supernatant. 200 µl was removed and added to the original cleared lysate. To 900 µl of samples, 100 µl of cold 100% TCA was added and the samples were left at 4°C overnight. TCA-precipitated material was pelleted 16,000xg (20 min, 4°C). The supernatant was removed by pipetting. Ice cold acetone (1 ml) was added and the tube was swirled. The sample was again centrifuged at 16,000xg (10 min 4°C), acetone was removed by pipetting, and pellet was dried 10 min. Pellets were resuspended in 6M urea/ 50 mM ammonium bicarbonate. Protein content of each sample was measured using Pierce BCA protein assay. 20 mM DTT was added 500 µg of protein and samples were incubated for 30 min at 37°C. Iodoacetamide was added at a final concentration of 50 mM and samples were incubated for 30 min in the dark at room temperature. Prior to trypsin digestion, urea concentration was diluted to less than 1M by adding water and pH adjusted to 8 with 1M Tris solution. 10 µg sequencing grade trypsin (Cat. No. V5280, Promega, Madison, WI) was added (1:50 enzyme to substrate ratio) and samples were incubated at 37°C with shaking for 16 hr. The reaction was stopped by addition of formic acid (FA) to a final concentration of 1% and the solution was desalted with a one cc (30 mg) Oasis HLB reverse phase cartridge (Cat. No. WAT054955, Waters, Milford, USA) conditioned with 3 × 500 µL acetonitrile (ACN), followed by 4 × 500 µL 0.1% FA. Samples were loaded onto the cartridges and washed with 3 × 500 µL 0.1% FA. Desalted peptides were eluted by 2 applications of 500 µL of 80% ACN/0.1% FA. Eluates were frozen, dried via vacuum centrifugation and stored at −80°C prior to peptide fractionation. For each sample the desalted peptides were labeled with iTRAQ 4-plex reagents according to the manufacturer's instructions (AB Sciex, Foster City, CA).The differentially labeled peptides were mixed to form a single pooled sample and subsequently desalted on 500 mg SepPak columns. The pooled sample was then fractionated by reverse phase chromatography into 21 fractions and then analyzed by liquid chromatography - tandem mass spectrometry. Briefly, each of the 21 fractions were resuspended in 8 µL of 3% ACN/0.5% FA before analysis using a Q Exactive mass spectrometer coupled to an EASY-nLC 1000 UHPLC (Thermo Scientific). A PicoFrit column (New Objective, Woburn, MA), with an inner diameter of 75 µm and packed with 20 cm of ReproSil-Pur C18 1.9 µm particles, was directly interfaced to the Q Exactive instrument equipped with a custom nano-electrospray ionization source. Two µg of the peptide mixture from each of the 21 fractions were injected and separated by a 180 min gradient from 5–60% solvent B. MS/MS analysis settings for protein identification were as follows: one precursor MS scan at 70,000 resolution in profile mode was followed by data-dependent scans of the top 12 most abundant ions at low-resolution (17,500) in profile mode. Dynamic exclusion was enabled for a duration of 20 s. MS/MS spectra were collected with a normalized collision energy of 28 and an isolation width of 2.5 m/z. The extensive peptide fractionation coupled with in depth MS analysis allowed detection and identification of very low levels of peptides. All MS data were processed using Agilent Spectrum Mill MS Proteomics Workbench (Agilent Technologies, Palo Alto, USA) Rev B.04.00.120. To identity of

significantly regulated proteins we used the Limma package in the R environment to calculate moderated *t*-test *P* values corrected by the Benjamini-Hochberg method.

## Determination of KatG levels

Cultures of mc$^2$155, ribosomal mutants, and ribosomal allelic-exchange mutants inducibly expressing MSMEG_6384 were grown in the absence of antibiotic to OD600 0.8 before being diluted to 0.25 and subjected to 6 hr induction with the indicated concentration of anhydrotetracycline. The cultures were collected by centrifugation then washed twice in a full volume of iced PBS + 0.05% Tween 80, resuspended in 400 µl extraction buffer (50 mM Tris-HCl pH 7.5, 5 mM EDTA, 0.6% SDS, 1 mM PMSF), and bead beaten with 200 µl 0.1 mm Zirconia/Silica beads (BioSpec) for a total of 3 min with 1 min on ice between pulses. Protein extracts were quantified using a standard BCA Protein Assay (Pierce), and 12.8 µg of extract was loaded on 10% Mini-PROTEAN TGX precast gels (Bio-Rad) and run at 200 V; equal loading between samples was assessed on a gel run in parallel using Coomassie blue staining. Electrophoretic transfer of the protein bands to PVDF membrane (Immun-Blot PVDF, Bio-Rad) was accomplished at 100 V for 60 min on ice, followed by blocking in TBST with 5% powdered milk for 60 min at room temperature with agitation. All antibodies used were diluted into TBST containing 1% powdered milk, and all incubation steps were completed on a rocking platform. Membranes were incubated with primary antibody (1:5,000) overnight at 4°, followed by 3 × 10 min washing in TBST, and 3.5 hr incubation at room temperature with secondary antibody (1:10,000). Primary antibody was obtained through the NIH Biodefense and Emerging Infections Research Resources Repository, NIAID, NIH: Monoclonal Anti-*Mycobacterium tuberculosis* KatG (Gene Rv1908c), Clone IT-57 (CDA4), NR-13793 RRID:AB_2631999. Secondary HRP-labelled goat anti-mouse antibody was acquired from Santa Cruz Biotechnology, Cat# sc-2005 RRID:AB_631736. Blots were developed using Western Lightning Plus-ECL (PerkinElmer). Results were confirmed in a repeated experiment, and data used in the figures is indicative of the initial experiment only.

Inducible expression of KatG was achieved by introduction of a plasmid, piKatG, into the rplO-1 AX1G allelic exchange mutant by electroporation, since the original isolate already carried an incompatible plasmid for GFP expression. piKatG is a derivative of pUV15tetORm in which the *gfp* ORF is replaced with a PCR product containing the MSMEG_6834 ORF. Primers were MSMEG_6834_Pac1-F and MSMEG_6834_EcoRV-R, and the resulting product was introduced into pUV15tetORm after digestion of each with PacI and EcoRV. pUV15tetORm and its derivatives express TetR, allowing for anhydrotetracycline-inducible expression of genes in *Mycobacterium* from a strong promoter into which tet operator sites have been engineered.

## Acknowledgements

We thank W Thompson-Butler and Melissa Choi for technical assistance, S A Stanley, S Grant, and A Barczak for helpful discussions. We thank J Stokes for extremely helpful input on ribosome assembly and Keith Derbyshire for providing *M. smegmatis* mc$^2$6. BBK thanks the Massachusetts General Hospital ECOR and NYCT Heiser Postdoctoral Fellowships for support. This work was funded by a Doris Duke Clinical Scientist Developmental award (DTH), the Broad Institute Tuberculosis donor group, and the Pershing Square Foundation.

## Additional information

### Funding

| Funder | Grant reference number | Author |
| --- | --- | --- |
| Doris Duke Charitable Foundation | 2008046 | Deborah T Hung |
| Pershing Square Foundation | | Deborah T Hung |
| Broad Institute Tuberculosis Donor Group | | Deborah T Hung |

The funders had no role in study design, data collection and interpretation, or the decision to submit the work for publication.

## Author contributions

JEG, Conceptualization, Data curation, Formal analysis, Investigation, Visualization, Methodology, Writing—original draft, Writing—review and editing; BBK-M, Conceptualization, Data curation, Formal analysis, Investigation, Methodology, Writing—original draft; CNW, PBK, Conceptualization, Investigation, Methodology; MRS, NR, Investigation; TRI, Data curation, Formal analysis, Investigation; RA, JL, Data curation, Formal analysis, Investigation, Methodology; SF, Formal analysis, Investigation; JCS, Resources, Formal analysis; SAC, Resources, Methodology; DTH, Conceptualization, Formal analysis, Funding acquisition, Investigation, Methodology, Project administration, Writing—review and editing

## Author ORCIDs

James E Gomez, http://orcid.org/0000-0002-2640-7453
Melanie R Silvis, http://orcid.org/0000-0002-0732-9744
Deborah T Hung, http://orcid.org/0000-0003-4262-0673

## Additional files

### Supplementary files

• Supplementary file 1. Primers and Plasmids.

• Supplementary file 2. 227 genes showing significant regulation (padj <0.05) in all ribosomal mutants.

• Supplementary file 3. Comparison of transcripts and proteins regulated >2 fold in the rplO-1 mutant.

### Major datasets

The following dataset was generated:

| Author(s) | Year | Dataset title | Dataset URL | Database, license, and accessibility information |
|---|---|---|---|---|
| James E Gomez, Nikolai Renedo, Jonathan Livny, Deborah T Hung | 2017 | Mycobacterium smegmatis ribosomal mutant transcriptomes | https://www.ncbi.nlm.nih.gov/bioproject/?term=PRJNA375121 | Publicly available at NCBI BioProject (accession no: PRJNA375121) |

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
