## [Decision Letter]

Thank you for submitting your article "Ribosomal mutations promote the evolution of antibiotic resistance in a multidrug environment" for consideration by *eLife*. Your article has been favorably evaluated by Richard Losick (Senior Editor) and three reviewers, one of whom, Michael S. Gilmore (Reviewer #1), is a member of our Board of Reviewing Editors. The following individuals involved in review of your submission have also agreed to reveal their identity: Helena Boshoff (Reviewer #2) and William R. Jacobs Jr. (Reviewer #3).

The reviewers have discussed the reviews with one another and the Reviewing Editor has drafted this decision to help you prepare a revised submission.

The manuscript "Ribosomal mutations promote the evolution of antibiotic resistance in a multidrug environment" by Gomez et al. investigates the mechanism by which mutants of *Mycobacterium smegmatis*, as a model for *Mycobacterium Tuberculosis*, acquire resistance to the antibiotic ciprofloxacin upon long-term exposure at 1.2 X MIC. It is state-of-the-art, interesting, and provocative. The mutant selection is innovative, as the authors set up mini-cultures in 384 well plates seeded with 10 cells in each well. When the population in each well reaches 10^5^ cells per ml, ciprofloxacin is added. This design ensures that the mutants analyzed from different wells are not siblings. Of those, 19 mutants were screened for polymorphisms in *gyrA*, and interestingly no mutations mapped to this gene. Whole genome sequencing was then performed on the 19 independent mutants. Four mutations map to the repressor of the LfrA efflux pump. Of the 15 remaining mutants, 14 possessed mutations in ribosomal genes. The authors generate specific mutations in the parental strain, and indeed confirm the mutations mediate the phenotypes. Interestingly, these mutations confer resistance to numerous other antibiotics that have notably different mechanisms of action, with the greatest cross-resistances to isoniazid, detergent and heat stress. Not only are the results interesting, the approach is innovative taking advantage of modern high throughput methodologies and will become a blueprint for future studies.

In review of the manuscript, two major concerns were noted: For technical reasons, a potential misinterpretation of the data cannot be excluded. The parental strain is actually not wild type, but rather a mutant of *M. smegmatis* ATCC 607 that was first isolated in 1988 (Snapper et al. PNAS 85: 6987-6991). The efficient plasmid transformation phenotype of this strain relates to its being permissive for plasmid replication (Snapper et al. 1990 Mol. Microbiol. 11:1911-1919). Recently, it was demonstrated that this property maps to a gene encoding an SMC protein (Panas et al. 2014 PNAS 111:13264-13271). The precise mechanism by which SMC allows plasmid replication has yet to be elucidated, but in wild type strains it either prevents replication or plasmid segregation to daughter cells. The authors use a concentration of ciplofloxocin that may not be bactericidal (in the second paragraph of the subsection “Selection for Low Level Ciprofloxacin Resistance Identifies Ribosomal Mutants”, it is stated that ciprofloxacin has a relatively slow killing rate – the data should be included as a supplemental figure), and the surrogate measure of escape from killing is indirect and relies on plasmid replication to measure green florescent protein activity. Therefore, it is possible, perhaps likely, that ciprofloxicin at the level being used may be limiting plasmid replication in the cell rather than selecting for resistance to killing, and the mutations obtained relieve that inhibition. It is therefore important to distinguish between allowing the pAL5000 replicon to replicate and cell death (It is ambiguous whether the GFP tag is on a self-replicating plasmid or integrated. Loss of the plasmid from the cell would be scored as a dead cell in most of the assays. Even if an integrated plasmid was used, excision could be dependent on the topology of the DNA altered by fluoroquinolones. In the Methods, it is important to note whether the selecting antibiotic is included during the experiments.) Several approaches might be useful in resolving the ambiguity, including directly assessing killing; using the parent of mc^2^155, mc^2^6, to see if the mutations identified confer resistance to bactericidal killing, such as engineering the *rplO* and *rplJ* mutations in mc^2^6, etc.

A second major concern is that a mechanism that confers resistance to both ciprofloxacin and isoniazid or ethionomide simultaneously is difficult to imagine. The authors mention that the proteomics, conducted on a single ribosomal mutant, *rplO*, did not indicate increased mistranslation as being a driving factor for these differences. However, it is not clear that the technique would be sensitive enough to detect mistranslation differences that could still drive increases in drug resistance, as reported by for example Su et al. (2016) and Leng et al. (2015). The transcriptional profile for *rplF* and *rplO* mutants are almost identical, which is perhaps not totally surprising since both of these are 23S rRNA interacting in the aminoacyltransferase site. (Why *rplO* and *rplF* mutants were chosen for the focus of further investigation is not entirely clear. Are there data to suggest that the other mutations in the large subunit or small subunit would, or would not phenocopy these mutants? In the one example of evolved high-level resistance in a single well (Figure 5), the mutation evolves following a mutation to a ribosome-associated gene (*rrlB*), which is not one of the two classes selected for further characterization.) Epistasis experiments demonstrate that all of these ribosomal mutants show increased WhiB7 expression (a transcription factor that may increase antibiotic efflux), and to some extent decreased KatG expression – two separate mechanisms by which resistance to different classes of antibiotic may occur. WhiB7 has been explored quite a bit in the literature due to its role in conferring multi-drug resistance and its upregulation by various drugs, especially tetracyclines, has previously been reported (Burian et al., 2012). What is missing is determination of the precise mechanism by which these mutations cause the specific upregulation of WhiB7 expression. Knowing the mechanism would represent a major advance.

---

## [Author Response]

*[…] In review of the manuscript, two major concerns were noted: For technical reasons, a potential misinterpretation of the data cannot be excluded. The parental strain is actually not wild type, but rather a mutant of M. smegmatis ATCC 607 that was first isolated in 1988 (Snapper et al. PNAS 85: 6987-6991). The efficient plasmid transformation phenotype of this strain relates to its being permissive for plasmid replication (Snapper et al. 1990 Mol. Microbiol. 11:1911-1919). Recently, it was demonstrated that this property maps to a gene encoding an SMC protein (Panas et al. 2014 PNAS 111:13264-13271). The precise mechanism by which SMC allows plasmid replication has yet to be elucidated, but in wild type strains it either prevents replication or plasmid segregation to daughter cells. The authors use a concentration of ciplofloxocin that may not be bactericidal (in the second paragraph of the subsection “Selection for Low Level Ciprofloxacin Resistance Identifies Ribosomal Mutants”, it is stated that ciprofloxacin has a relatively slow killing rate – the data should be included as a supplemental figure), and the surrogate measure of escape from killing is indirect and relies on plasmid replication to measure green florescent protein activity. Therefore, it is possible, perhaps likely, that ciprofloxicin at the level being used may be limiting plasmid replication in the cell rather than selecting for resistance to killing, and the mutations obtained relieve that inhibition. It is therefore important to distinguish between allowing the pAL5000 replicon to replicate and cell death (It is ambiguous whether the GFP tag is on a self-replicating plasmid or integrated. Loss of the plasmid from the cell would be scored as a dead cell in most of the assays. Even if an integrated plasmid was used, excision could be dependent on the topology of the DNA altered by fluoroquinolones. In the Methods, it is important to note whether the selecting antibiotic is included during the experiments.) Several approaches might be useful in resolving the ambiguity, including directly assessing killing; using the parent of mc^2^155, mc^2^6, to see if the mutations identified confer resistance to bactericidal killing, such as engineering the rplO and rplJ mutations in mc^2^6, etc.*

Major point 1: potential issues surrounding the use of mc^2^155.

The first major concern related to the genetic background of the strain used for all of our studies, *M. smegmatis* mc^2^155. We now include data showing that indeed, the phenomenon we describe for mc^2^155 also applies to the parental mc^2^6 strain and is thus not an artifact of the genetics of mc^2^155. Specifically, we have addressed this issue on several levels:

1) While the emergence of the resistant mutants were detected based on GFP fluorescence, microtiter plate-based MICs in several instances were measured by optical density, which mitigates concerns that the “resistance” observed in these mutants is due simply to altered (GFP) plasmid retention.

2) Nonetheless, we have now obtained mc^2^6 (parental strain) and used this strain in a de novo selection experiment to show that CIP-resistant ribosomal mutants could arise spontaneously in this true WT background in the same manner as in mc^2^155.

3) Finally, we have now repeated the microtiter-plate based selection process with the original mc^2^155 strain carrying the episomal GFP plasmid and with mc^2^6 and found that there is no significant plasmid loss. Both of these selections were conducted in the absence of hygromycin and survival enumerated in the presence and absence of hygromycin (described in detail in Figure 1—figure supplement 2) thereby enabling us to directly assess the bactericidal effect of ciprofloxacin and the effects of ciprofloxacin stress on plasmid retention in the mc^2^155 strain. There is no significant plasmid loss. Most importantly, we identified two *rplO* mutants that arose from the mc^2^6 background during CIP selection, and both of these were also cross-resistant to INH showing that the same phenomenon that we have described for mc^2^155 holds true for mc^2^6.

A second major concern is that a mechanism that confers resistance to both ciprofloxacin and isoniazid or ethionomide simultaneously is difficult to imagine. The authors mention that the proteomics, conducted on a single ribosomal mutant, rplO, did not indicate increased mistranslation as being a driving factor for these differences. However, it is not clear that the technique would be sensitive enough to detect mistranslation differences that could still drive increases in drug resistance, as reported by for example Su et al. (2016) and Leng et al. (2015). The transcriptional profile for rplF and rplO mutants are almost identical, which is perhaps not totally surprising since both of these are 23S rRNA interacting in the aminoacyltransferase site. (Why rplO and rplF mutants were chosen for the focus of further investigation is not entirely clear. Are there data to suggest that the other mutations in the large subunit or small subunit would, or would not phenocopy these mutants? In the one example of evolved high-level resistance in a single well (Figure 5), the mutation evolves following a mutation to a ribosome-associated gene (rrlB), which is not one of the two classes selected for further characterization.) Epistasis experiments demonstrate that all of these ribosomal mutants show increased WhiB7 expression (a transcription factor that may increase antibiotic efflux), and to some extent decreased KatG expression – two separate mechanisms by which resistance to different classes of antibiotic may occur. WhiB7 has been explored quite a bit in the literature due to its role in conferring multi-drug resistance and its upregulation by various drugs, especially tetracyclines, has previously been reported (Burian et al., 2012). What is missing is determination of the precise mechanism by which these mutations cause the specific upregulation of WhiB7 expression. Knowing the mechanism would represent a major advance.

Major point 2: issues surrounding a common resistance mechanism to multiple drugs (ciprofloxacin and isoniazid).

Indeed, the major finding of this manuscript that we wish to report is both surprising and remarkable, that a single mutation can result in resistance to antibiotics with different mechanisms of action. We apologize if our discussion of the data providing insight into the biology and mechanism underlying this phenomenon was not clear in the original manuscript, thereby preventing the reviewers from understanding what we believe is going on. We have tried to clarify this in the current draft.

1) Based on transcriptomic and proteomic analysis of the mutants, we conclude that the perturbations in ribosomal function due to these mutations leads to a large-scale reorganization of the transcriptome and the proteome. Changes in specific genes within the large-scale reorganization can be used to account for the individual resistance phenotypes that we see to individual antibiotics. For example, a repression of KatG expression accounts for the observed resistance to isoniazid while the upregulation of WhiB7 contributes to ciprofloxacin resistance. We have tried to clarify that changes in specific genes within the global shifts can account for resistance to different antibiotics (subsection “Genome-Scale Studies suggest an overall transcriptional reprogramming contributing to multi-drug resistance”, first two paragraphs). In addition, we now provide additional data to suggest that the global changes that occur in the ribosomal mutants may be occur as a result of the ribosomal mutations causing stress to the bacterial cell due to defective assembly of the large 70S ribosome, thereby providing further molecular insight into the consequences of carrying these described ribosomal mutations (Figure 4, subsection “Genome-Scale Studies Illuminate the effects of ribosomal mutations on ribosome assembly”, first three paragraphs).

2) The question of the mutations causing mistranslation as a cause of resistance was raised. Several lines of evidence argue against mistranslation as the underlying cause of increased resistance to antimicrobials.

In *M. smegmatis*, mistranslation has been clearly linked to rifampin resistance; we, however, see no shift in sensitivity to rifampin.

Deletion of *whiB7* eliminates the resistance to several of the antibiotics, which should *not* occur if mistranslated targets are the source of resistance.

The question was raised whether the proteomic data was sensitive enough to detect mistranslated peptides. We now have clarified in the last paragraph of the subsection “Genome-Scale Studies Illuminate the effects of ribosomal mutations on ribosome assembly”, that the method is indeed sensitive enough as we are able to detect a significant number of mistranslated peptides even in wild-type. Importantly, there are *no* differences in numbers of mistranslated peptides between the ribosomal mutant and its parent. Therefore, mistranslation cannot account for the multidrug phenotype.

3) The concern was raised whether we could truly infer that a common phenomenon was occurring across all of the ribosomal mutants that was leading to the shared resistance phenotypes as we had only characterized one *rplO* and one *rplF* mutant. To address this, we have expanded our unbiased transcriptomic analysis to include mutants in all loci identified (9 mutants in total), including multiple *rplO, rpsE*, and 23S rRNA mutants, along with mc^2^155 and an *lfrR* mutant for comparison. This new, rich dataset shows that all of the mutants are behaving very similarly based on a common transcriptional program and now allows us to identify transcriptional perturbations that are common across all ribosomal mutants sharing these resistance phenotypes, that explain some of the individual resistance mechanisms (i.e., downregulation of katG to explain isoniazid resistance and WhiB7 induction to explain ciprofloxacin resistance). This new data strongly supports the notion that a common, highly altered transcriptional program relative to wild-type underlies the resistances observed in this manuscript.